# Parameters vs FLOPs: Scaling Laws for Optimal Sparsity for Mixture-of-Experts Language Models

Samira Abnar [1] [*]   Harshay Shah [2] [*]   Dan Busbridge [1]   Alaaeldin El-Nouby [1]   Joshua M Susskind [1]
Vimal Thilak [1] [*]

## Abstract

Scaling the capacity of language models has consistently proven to be a reliable approach for improving performance and unlocking new capabilities. Capacity can be primarily defined by two dimensions: the number of model parameters and the compute per example. While scaling typically involves increasing both, the precise interplay between these factors and their combined contribution to overall capacity remains not fully understood. We explore this relationship in the context of sparse Mixture-of-Experts (MoEs), which allow scaling the number of parameters without proportionally increasing the FLOPs per example. We investigate how varying the sparsity level, i.e., the fraction of inactive parameters, impacts model's performance during pretraining and downstream few-shot evaluation. We find that under different constraints (e.g., parameter size and total training compute), there is an optimal level of sparsity that improves both training efficiency and model performance. These results provide a better understanding of the impact of sparsity in scaling laws for MoEs and complement existing works in this area, offering insights for designing more efficient architectures.

## 1. Introduction

Empirical scaling laws for language model pretraining (Kaplan et al., 2020; Hoffmann et al., 2022; OpenAI, 2023; 2024; Gemini Team et al., 2024; Henighan et al., 2020; Clark et al., 2022; Yun et al., 2024; Ludziejewski et al., 2024) have demonstrated that proportionally increasing model capacity, along with data and total compute budget, consistently decreases pretraining loss (i.e., perplexity), improves downstream task performance (Devlin et al., 2019; Brown et al., 2020; BIG-bench authors, 2023) and unlocks emergent capabilities (Wei et al., 2022a).

A recurring notion in these studies is that model capacity is well quantified by the total number of model parameters. However, the number of parameters is not the only means to increase model capacity. *Compute per example (i.e., a fixed-sized input)*, measured in FLoating OPerations (FLOPs), also plays a significant role (Clark et al., 2022). In fact, several mechanisms (Shazeer et al., 2017; Dehghani et al., 2019; Wei et al., 2022b; Goyal et al., 2024; Csord'as et al., 2024) allow for independent variation of the number of parameters or FLOPs per example within a model. For instance, Sparse Mixture-of-Experts (MoE) models (Shazeer et al., 2017) introduce "FLOP-free parameters" by leveraging sparsity, where only a subset of expert modules is activated for each input.

When studying scaling laws for specific classes of models, e.g., vanilla transformers, the total number of parameters can serve as a reasonable relative estimator of FLOPs per example. Therefore, using the number of parameters as a measure of model capacity in scaling law studies is appropriate. In scenarios or for architectures where the number of parameters and FLOPs per example are not directly linked, it is essential to jointly consider the effects of these variables on scaling model capacity (Clark et al., 2022). We therefore ask

> *"Can we draw scaling laws for the optimal trade-off between*
> *parameter count and FLOPs per example?"*

To address this question, we study sparse Mixture-of-Expert Transformers (MoEs) (Shazeer et al., 2017; Lepikhin et al., 2021; Fedus et al., 2022; Zoph et al., 2022; Muennighoff et al., 2024) in the context of language modeling. Existing scaling law studies for MoEs, investigate the role of variables like number and granularity (Ludziejewski et al., 2024) of experts, underlying dense model size and inference compute in predicting the performance of the models under different conditions such as training or inference compute optimality (Du et al., 2021; Clark et al., 2022; Yun et al.,

---

[*]Core Contributors   [1]Apple [2]MIT. Correspondence to: Samira Abnar <abnar@apple.com>, Vimal Thilak <vthilak@apple.com>.

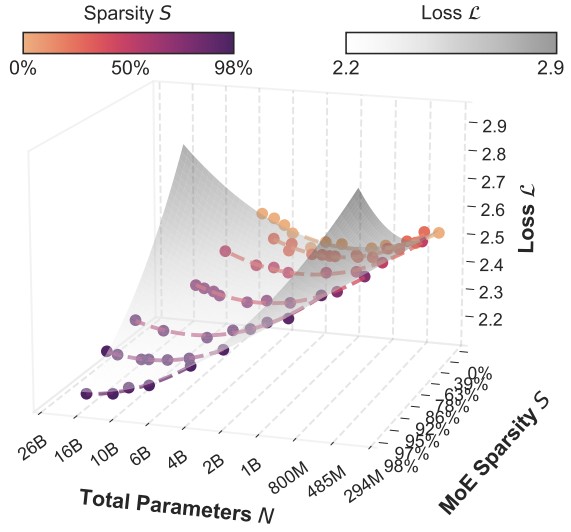

(a) IsoFLOP surface over sparsity and total parameters

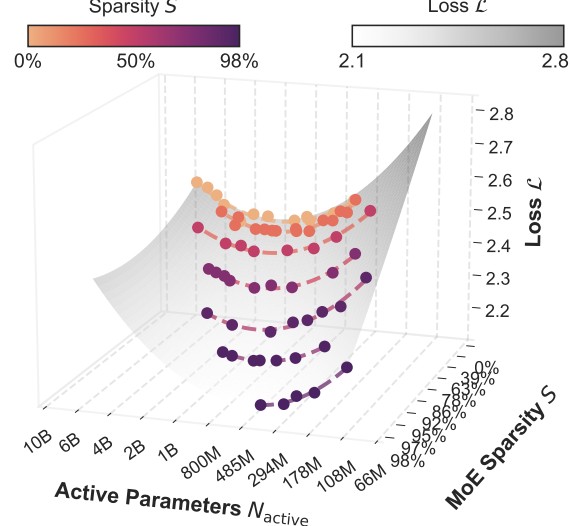

(b) IsoFLOP surface over sparsity and active parameters

*Figure 1.* **IsoFLOP surface over observed pretraining loss L, model size (in terms of total N and active parameters $N_a$), and sparsity S.** We fit a polynomial function mapping $N$ (or $N_a$), $S$, and their interaction to $L$, using empirical data. For both fits the MSE loss for predicting loss on a held out set is 0.0001. These results indicate that for a fixed compute budget, increasing model sparsity leads to a reduction in pretraining loss. When considering optimal model size, we observe opposite trends for total parameters ($N$) (Figure a) versus active parameters ($N_a$) (Figure b). (See Figure 8 in Appendix D.1 for results with different total compute budgets $C$.)

2024; Ludziejewski et al., 2024). In this paper, we focus on the interaction between FLOPs per example and total parameter count, and their impact on model performance in MoEs, through a large-scale empirical study.

We define sparsity as the ratio of inactive experts to the total number of experts, which controls the ratio of the total number of parameters to FLOPs per example in MoEs. We evaluate loss and downstream metrics for different sparsities, model sizes, and compute budgets. Through qualitative and quantitative analysis to derive scaling laws which disentangle total parameters vs FLOPs per example in MoEs, we can estimate the optimal sparsity level under the setting where both total training FLOPs and total number of parameters are given and fixed. Generally, we find that:

- During pretraining, increasing a model's capacity by adding more parameters yields greater benefits than increasing FLOPs per example. We observe that the size of compute-optimal models increases as we increase the training budget (measured in terms of total FLOPs) while the active number of parameters, hence FLOPs per example, decrease for compute-optimal models.

- During inference, FLOPs per example seem to play a more important role[1]. For many tasks, upstream performance is a good predictor of downstream per-

formance and the relationship between upstream and downstream performance is not impacted by the sparsity level. However, on downstream tasks that presumably require more "reasoning", we observe that for models with the same perplexity on the pretraining data distribution, sparser models, i.e., models with fewer active parameters, perform worse.

Our results, in line with findings from previous relevant studies (Ludziejewski et al., 2024; He, 2024) on scaling laws for MoEs, show increasing sparsity level leads to better performance and efficiency during pretraining. Considering the various methods to increase compute per example during inference adaptively conditioned on task or example complexity, we conclude that approaches like MoEs, which reduce the unit compute cost (i.e., FLOPs per token) by increasing the sparsity level, hold significant promise given their potential to enhance efficiency in both pretraining and inference.

## 2. The Interplay between Model Parameters and Sparsity in MoEs

Is there an optimal trade-off between parameter count and FLOPs per example in MoEs under the setting where the training compute budget (i.e., total training FLOPs) is fixed?

---

[1]A relevant discussion here is the recent trend of increasing test-time compute, e.g., OpenAI o1 model (OpenAI, 2024), achieved

by generating more tokens as a way for introducing parameter-free-FLOPs.

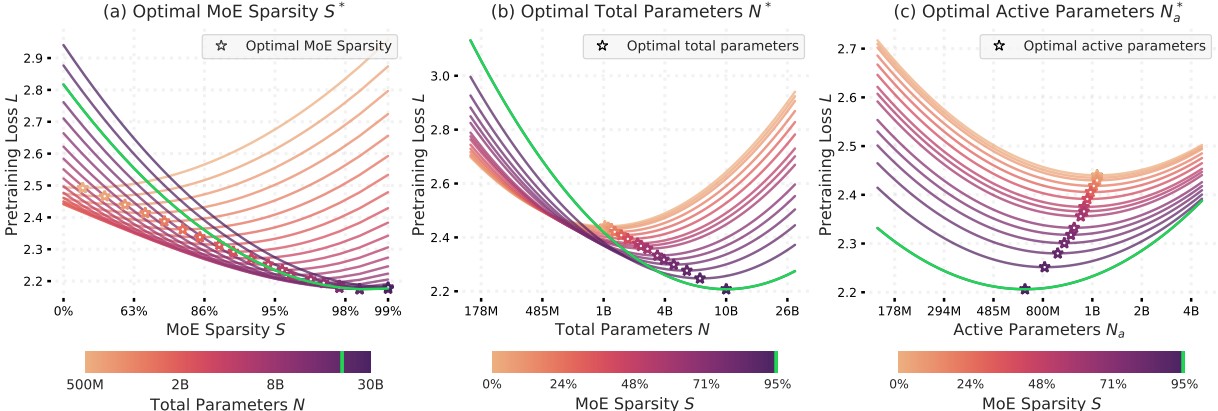

*Figure 2.* **IsoFLOP slices along Sparsity and Model Size** ($C = 1e20$). We use fitted isoFLOP surfaces (Section 2) to analyze how sparsity **S** and model size **N** impact the loss **L** for a fixed compute budget. We identify optimal points by (a) fixing **N** and varying **S**, (b) fixing **S** and varying **N** and (c) fixing **S** and varying active parameters **$N_a$**. Observe that (a) the optimal sparsity $S$ increases with increasing model size $N$ and converges to 1 while (b) and (c) show that the optimal model size $N$ and active parameter count $N_a$ increase and decrease respectively with increasing sparsity levels. (see Figure 9 in Appendix D.1 for other total training compute budgets.)

Intuitively, under infinite data setting, scaling model capacity along with the training compute budget leads to performance improvements. Previous scaling law studies suggest that, conditioned on a training compute budget measured in FLOPs denoted by $C$, the optimal number of parameters, $N^*(C)$, exhibits a power-law relationship with $C$ (Hoffmann et al., 2022):

$$N^*(C) = \arg \min_N \mathcal{L}(N; C) \propto C^a \qquad (1)$$

Our goal is to study how to optimally trade-off FLOPs per example and total parameters in MoEs. In MoEs the balance between parameters and FLOPs can be expressed through the sparsity level, $S$. We define $S$ as the ratio of non-active to total number of experts, i.e., $S = \frac{E-K}{E}$; where $E$ is the total number of experts and $K$ is the number of selected experts per token. We can vary the sparsity level by either changing the number of active experts $K$ or total number of experts $E$. [2] Essentially, for models with the same $N$, the model with a higher $S$ will have fewer active parameters $N_a$, resulting in fewer FLOPs per example. For more details on the notations and experimental settings see Appendix A and Appendix B.

$$(N^*, S^*) = \arg \min_{N,S} \mathcal{L}(N, S; C) \qquad (2)$$

To simplify the problem of understanding the joint role of $N$ and $S$ in predicting $L$, we break the problem, Equation 2,

---

[2]Sparsity level determines the number of active parameters given the total number of parameters and we use the active number of parameters as a proxy for FLOPs per example, as $6N_aD$ provides a good estimate of the total FLOP count for MoEs; see Appendix C for details.

into two parts:

1. *"How does the sparsity level impact the scaling laws of the relationship between $N$ and $C$ for training-compute optimal models?"* To address this question in §2.1, we fix $S$ and vary $N$, studying how optimal $N$ and $N_a$ change for different values of $S$:

$$N^* = \arg \min_N \mathcal{L}(N; C, S) \qquad (3)$$

2. *"Is there an optimal balance between total number of parameters and the sparsity level under fixed training-compute budget?"* To address this question in §2.2, we fix $N$ and vary $S$, studying how optimal $S$ changes across different values of $N$:

$$S^* = \arg \min_S \mathcal{L}(S; C, N) \qquad (4)$$

As the first step, considering a fixed training compute budget $C$, we fit a 3D surface, referred to as the IsoFLOP surface, in Figure 1a, using a polynomial function, following approach II of Hoffmann et al. (2022). Compared to Hoffmann et al. (2022) we include the sparsity variable and fit a single 3d IsoFLOP surface across all data points, rather than fitting separate 2d IsoFLOP curves for fixed sparsity levels or model sizes. We conducted a grid search to determine the optimal polynomial degree for $N$, $S$, and the interaction term $N \times S$, finding that a degree of $(2, 2, 2)$ resulted in the lowest cross-validation error. Both $N$ and $S$ are in log space (see Appendix B for more details).

As seen in Figure 1a, the IsoFLOP surface plot is parabolic along model size, suggesting that the findings of Hoffmann

et al. (2022) extend to MoEs across different sparsity levels, i.e., $\mathcal{L}(N; C, S)$ is parabolic, with its optimal solution located at the turning point. When considering the total number of parameters $N$, the optimal value increases as the sparsity level increases, while for the active number of parameters $N_a$ the optimal value decreases with the sparsity level. This indicates that by increasing the sparsity level the training compute optimal models are larger but have fewer FLOPs per example, i.e., lower inference cost. Moreover, along sparsity, the pretraining loss decreases monotonically, indicating that, for the same compute budget, sparser models achieve better pretraining performance. We observe the same pattern across different training compute budgets (See Appendix D.1). To better understand and explain these observations, we examine slices of the IsoFLOP surface along the axes of $S$ and $N$ separately in §2.1 and §2.2, respectively.

## 2.1. Optimal Model Size for Fixed Sparsity Level

Here we examine how sparsity influences scaling laws governing the relationship between $N$, $N_a$ and $C$ for training-compute optimal models, i.e. how does $N^*$ and $N_a^*$, for a given $C, S$ (Equation 3), change as we increase $S$? Looking at slices of the IsoFLOP surface along the model size dimension, in Figure 2b and Figure 2c, we observe how the IsoFLOP curves shift along loss and model size. Considering the training-compute optimal model, for a fixed compute budget, loss decreases as we increase sparsity. Furthermore, while sparser models have larger $N$ compared to denser models, as seen in Figure 2b, they have a smaller active parameter count $N_a$; hence, fewer FLOPs per example. Intuitively, more parameters in total increase the capacity of the sparser models to fit the data, while fewer number of active parametes, hence fewer FLOPs per example, allow the model to be trained with more tokens, i.e., higher $D$, for the same training compute budget.

## 2.2. Optimal Sparsity Level for Fixed Model Size

In this section we aim to understand the dynamics between the total number of parameters and FLOPs per example in MoEs. In Section 2.1 we are considering the case where there is no bound on the total number of parameters. In this case, we observe that under fixed training compute budget in terms of FLOPs, it is better to train sparser models with higher total number of parameters. However in practical scenarios it is reasonable to assume that there would be some bounds on the memory and hence the total number of parameters of a model. This leads us to a fundamental question: Is there an optimal balance between the total number of parameters and and FLOPs per example under a fixed training-compute budget? Thus, we investigate the optimal sparsity level when total number of parameters is fixed. Specifically, we ask: Given $N$ and $C$, How does $S^*$ change as we vary $N$?

To address this, we look into slices of the IsoFLOP surface along the sparsity dimension. As we can see in Figure 2a, for a fixed training compute budget and fixed model size $\mathcal{L}(S; N, C)$ exhibits a parabolic profile, reaching its optimum value at the vertex where $S = S^*$. It is noteworthy that for a given total training compute, there is threshold value $N_{th}$ for the total number of parameters, where for larger models, models with $N > N_{th}$, increasing sparsity always has a positive impact, i.e., optimal sparsity level approaches 1.0. More accurately, for a fixed compute budget the optimal sparsity level increases with model size and converges to 1 as the model size grows (see Figure 4 in §D.2 in the Appendix for more details). Note that the optimal model, here is not the largest model, i.e., there is a compute optimal model size in terms of total parameters even after sparsity is introduced, and increasing total number of parameters would lead to under-training if training compute budget is fixed.

These results highlight the importance of balancing the number of parameters with FLOPs per example in MoEs. Intuitively, when the total number of parameters is small, higher sparsity results in fewer active parameters, and thus fewer FLOPs per example. This reduction in FLOPs per example may lead to inefficiencies during both training and inference. Conversely, when the total number of parameters is large, for a reasonable amount of FLOPs per example, a fixed compute budget may not allow sufficient training on enough tokens to make use of the model's additional capacity.

## 3. Impact of Training Compute Budget on the Interaction between Model Parameters and Sparsity

Does increasing compute budget impact the interaction between the parameters and FLOPs per example in MoEs and how they contribute to model's capacity? In other words, does the recipe for optimally increasing model capacity, i.e., optimal sparsity level for MoEs change as we scale up the total training compute?

To answer this question. in Figure 3 we illustrate the trends for changing the total number of parameters, $N^*$, the number of active parameters, $N_a^*$, and the loss, $L^*$, with sparsity level across different compute budgets.

Figure 3c shows that the optimal sparsity level approaches 1 across all compute budgets used in our experiments. There is no significant difference observed in the slope of the loss vs sparsity curves across different training compute budgets used in our experiments. This observation suggests that there is no diminishing effect of sparsity on the pretraining loss as we increase training compute budget, i.e., if there is no constraint on the model size, sparsity improves the

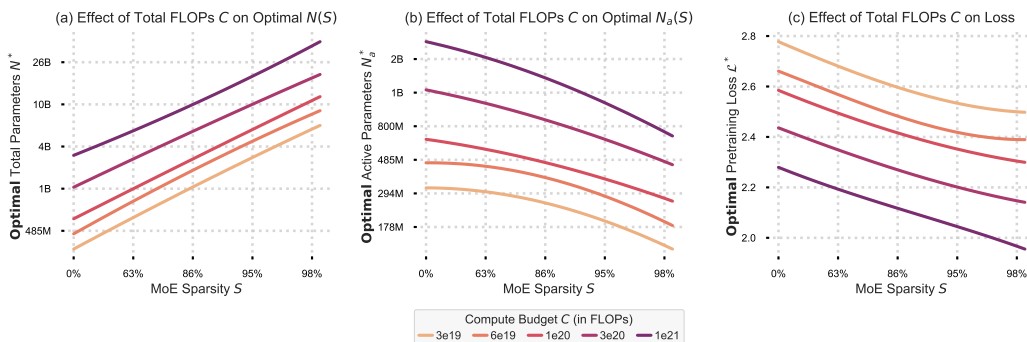

*Figure 3.* **Effect of compute budget on model size, number of active parameters and loss with sparsity.** Across all compute budgets, we observe that (a) the optimal model size $N$ increases with sparsity, (b) the optimal number of active parameters $N_a$ decreases with sparsity, and (c) the loss $L$ decreases with sparsity.

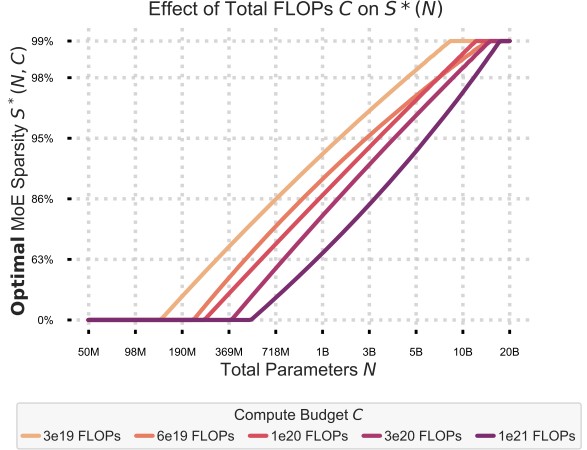

*Figure 4.* **Effect of training budget $C$ and total parameters $N$ on MoE sparsity.** Optimal MoE sparsity $S^*$ changes with respect to the total number of parameters $N$ and the training budget $C$. The $x$-axis represents the total parameters $N$ on a logarithmic scale, and the $y$-axis shows the optimal MoE sparsity $S^*$.

performance of the model across all training budgets.

In Figure 3a and Figure 3b, , we see a consistent trend of increasing $N$ and decreasing $N_a$ for compute optimal models as sparsity level increases across all training compute budgets. Moreover, as can be seen in Figure 4, when model size in terms of total number of parameters is fixed, optimal sparsity level decreases with training compute budget while increases with model size as discussed in Section 2.2.

## 4. Effect of MoE Sparsity on Downstream Task Performance

In this section, we study how sparsity affects the relationship between upstream and downstream performance of MoEs. In other words, does sparsity impact the relative gains from improvements in pretraining tasks on downstream tasks?

We use downstream tasks from the evaluation suite in `llm-foundry`[3] for benchmarking our pretrained models, specifically in an in-context few-shot learning setup. This setup focuses on evaluating a model's ability to learn and adapt to new tasks with limited examples. The downstream task are devided into four pre-defined categories namely: language understanding, world knowledge, reading comprehension, and symbolic reasoning to help us systematically test whether the downstream vs upstream performance trend remains the same or is different as we vary sparsity values.

We observe from Figure 5a (language understanding), Figure 5c (commonsense reasoning), and Figure 5d (world knowledge) that, in an in-context few-shot learning setting, there is a strong correlation between upstream (pretraining) loss and downstream performance (error) across all these tasks. For these tasks, downstream performance in the few-shot setting is predictable based on upstream performance, regardless of the sparsity level. This indicates that, in the context of these tasks, the optimal sparsity level follows the same trend as the optimal sparsity observed during pretraining. However, Figure 5b (reading comprehension) shows an example of a task where models with higher sparsity transfer more poorly compared to denser models. This decrease in the transfer performance of sparser models on these tasks may be due to the lower inference-time compute in sparser models compared to their denser counterparts for a similar pretraining loss. Further analysis is needed to verify this intuition.

If fewer FLOPs per example are the reason behind the worse transfer performance in sparser models, this effect might diminish with a larger total training compute budget, as the optimal active number of parameters increases. Moreover, one can use approaches like chain-of-thought reasoning (Wei et al., 2022b) to independently increase FLOPs per example

---

[3]Github repository: `https://github.com/mosaicml/llm-foundry`

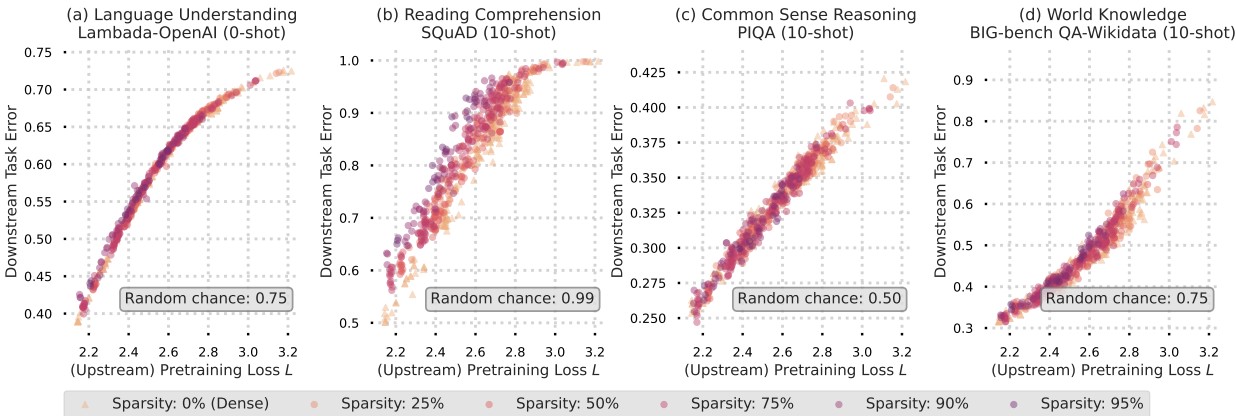

*Figure 5.* **Effect of sparsity on downstream vs upstream performance.** Downstream error shows a tight relationship with pretraining ("upstream") loss across downstream tasks across all sparsity levels.

during inference time.

In Appendix E, we explore whether increasing inference-time compute via Chain-of-Thought (CoT) prompting can improve the performance of MoEs on tasks that require more reasoning. Our experiments indicate that MoEs benefit more from this increased compute compared to dense models with a similar number of active parameters. This suggests that dynamic compute allocation during inference may be crucial for MoEs to perform well on complex reasoning tasks.

While our results may indicate that there may be no additional benefit obtained via sparsity in MoEs beyond the efficiency gains for pretraining, we caution the reader that this suggestion may be an artifact of the scale of our experiments. In the end, since, as shown in §2, sparser models are more efficient both in terms of training and inference cost (when measured in terms of theoretical FLOPs), we can reach better pretraining performance with higher sparsity levels at a lower cost, which can translate to better downstream performance.

## 5. Incorporating Sparsity into Scaling Laws

The scaling laws proposed by Kaplan et al. (2020) provide a framework for predicting loss in dense models by establishing a power-law relationship between loss $L$, number of parameters $N$ and dataset size $D$, where $N$ and $D$ interact linearly. Formally, the relationship is given by:

$$L(N, D) = \frac{a}{N^\alpha} + \frac{b}{D^\beta} + e \tag{5}$$

Here, the term $N^\alpha$ captures the inverse relationship between model size and loss, where an increase in model size $N$ leads to a reduction in loss. The exponent $\alpha$ quantifies the rate of this decrease; a larger $\alpha$ suggests a steeper reduction in loss with increasing model size. Similarly, the term $D^\beta$ indicates the impact of dataset size $D$ on loss, with larger datasets contributing to lower loss values. The exponent $\beta$

measures this relationship, where a larger $\beta$ implies a greater benefit from increased data. The constant $e$ represents an asymptotic minimum for the loss, as both model size and dataset size approach infinity.

For dense models with a fixed total training FLOPs, $C$, the parameters $N$ and $D$ are interrelated through the equation for estimating FLOPs per example, given as $C = 6ND$ for transformers. However, in MoEs (Mixture of Experts models), this relationship involves the active number of parameters $N_a$ rather than the total parameter count $N$. Thus, $D$ and $N_a$ define the total training FLOPs rather than $D$ and $N$. Given the analysis conducted in §2, we know that if the total number of parameters $N$ is fixed, the optimal sparsity level, i.e., active number of parameters would depend on $N$. Motivated by this observation, we suggest the following parametric form that includes a multiplicative interaction between $N$ and $S$ or $N_a$ to predict the loss:

$$L(N, D, S) = \frac{a}{N^\alpha} + \frac{b}{D^\beta} + \frac{c}{(1-S)^\lambda} + \frac{d}{(1-S)^\delta N^\gamma} + e \tag{6}$$

The term $(1 - S)$ in the above equation provides a rough estimate of the percentage of active parameters. If the exponent for the multiplicative terms is the same then that term provides an approximate estimate of the number of active parameters.

By incorporating sparsity into the scaling law equation, we can eliminate the need for parameters specific to MoEs, such as the total and active number of experts. As demonstrated by Frantar et al. (2024), this formulation also holds for other sparsity mechanisms, such as weight sparsity, where individual neural network connections are pruned.

We use the recipe described by Hoffmann et al. (2022) and use the L-BFGS algorithm to fit the coefficients in equation 6 using a Huber loss with $\delta = 10^{-3}$. Optimal coefficient values were determined through a grid search (see

Table 2 for search values). The results of data fitting and validation are shown in Figure 6. The estimated values are shown in Table 3 in Appendix F.

## 6. Discussion

Our findings amplify the findings of Ludziejewski et al. (2024) and further justify the effort to work toward MoEs with experts larger in number and smaller in size (He, 2024). For downstream tasks which their performance is predictable given the pretraining loss (i.e., perplexity), sparsity potentially provides efficiency gains both during pretraining and inference.

Here is a summary of our observations as discussed in Sections 2 to 5 :

- **Larger, Sparser Models Perform Better under a Fixed Compute Budget:** When memory and communication overheads are disregarded, increasing sparsity while proportionally expanding the total number of parameters consistently leads to a lower pretraining loss, even when constrained by a fixed training compute budget (see § 2).

- **Optimal Sparsity for Fixed Model Size:** For any given number of parameters and under a fixed training compute budget, model performance as a function of sparsity exhibits a parabolic pattern, reaching its peak at an optimal sparsity level (see §2.2). Specifically, the optimal sparsity level:

  - Increases with the total number of parameters approaching 1.0 for larger models. i.e., if a model is relatively small for a given training compute budget, sparsifying it more than a threshold will hurt its performance. On the other hand, if a model is relatively large for a given compute budget, further sparsifying it helps as it leads to increase in the number of tokens the model is trained on under the given training budget constraints (see §2.2).

  - Increases across all model sizes as the training compute budget increases (see §D.1 and §D.2).

- **Effect of Sparsity on Scaling Laws for Optimal Model Size:** For any specific sparsity level, performance of the models as a function of their size exhibits parabolic behavior under a fixed training compute budget. i.e., the model reaches its optimal performance at a vertex, that indicates optimal model size. Under these conditions:

  - The optimal active number of parameters decreases as the sparsity level increases, leading to smaller FLOPs per example and more efficient inference even though the total number of parameters increases (see §2.1).

  - While the trend of increasing active number of parameters is similar across all training compute budgets; the optimal active number of parameters decrease more rapidly with sparsity as the training compute budget increases (see §3).

- **Effect of Sparsity on Downstream Performance:** For most downstream tasks, models with similar pretraining perplexity have similar downstream task performance regardless of sparsity. For reading comprehension tasks (e.g., CoQA (Reddy et al., 2019), SQuAD (Rajpurkar et al., 2018)), denser models perform better, potentially due to their higher inference-time compute than a perplexity-matched sparse model. Strategies to increase inference time compute dynamically (Wei et al., 2022b; Goyal et al., 2024) may address this gap.

- **Parametric Scaling Law:** We propose a parametric form for scaling laws that accounts for sparsity. The model coefficients are estimated using the empirical data obtained by training compute-optimal models. An interesting observation from Appendix F is that the exponent for sparsity term $\lambda$ is negative which is consistent with our intuition that sparser models lead to a lower perplexity.

### 6.1. Limitations

In our analysis, similar to other scaling law studies (Kaplan et al., 2020; Hoffmann et al., 2022), we have measured the costs for both training and inference exclusively in terms of FLOPs. While there may be discrepancies between actual computational costs and theoretical FLOPs due to hardware specifications, infrastructure, and implementation details, it is reasonable to abstract away from these factors when comparing similar models under fixed conditions. However, an important aspect not accounted for in this study is the cost associated with memory usage and communication overhead, which could potentially increase as we raise the sparsity level. Incorporating these factors is challenging because they are highly dependent on the hardware used. To address this limitation to some extent, in Section 2.2 we investigate the optimal sparsity level under the setting where total number of parameters is fixed.

Despite the limitation with using an approximate method to quantify FLOPs, our findings highlight the importance of investing in methods to enhance the efficiency of sparse Mixture-of-Experts models. By increasing model capacity through additional parameters while minimizing per-unit computation costs, these models have the potential to improve both efficiency and performance. The availability of GPUs with larger memory, for e.g., the recently introduced H200 GPU chip with 141 GB of memory as well as improving the efficiency of training and deployment pipelines (NeMo Authors, 2025) suggest that there is significant interest in developing efficient implementations for

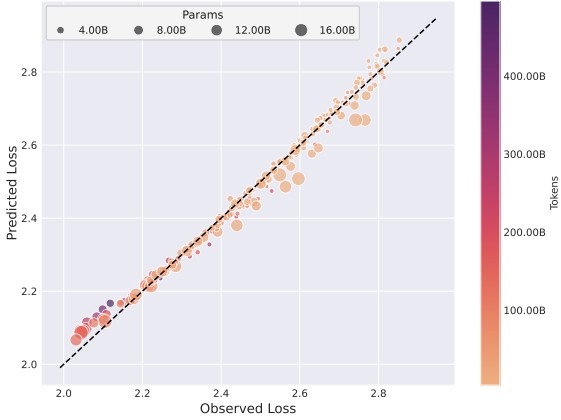

(a) Fit on data used to estimate coefficients.

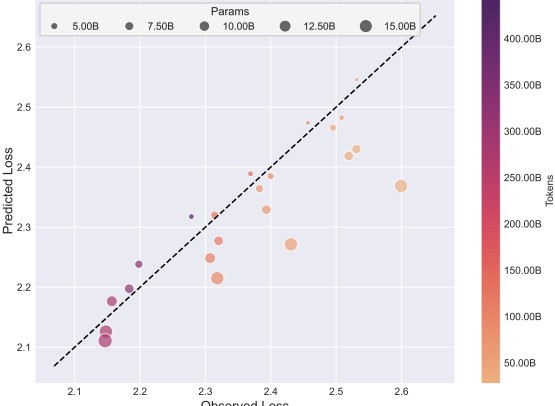

(b) Validating scaling law on held-out dataset.

*Figure 6.* Scaling law fit on data obtained from training compute-optimal models. Figure 6(a) shows the fit on the data used to estimate the coefficients for equation 6, while Figure 6(b) validates these coefficients on a held-out dataset. All data points with $S = 0.98$ were excluded from the fitting process for out-of-sample validation. The dashed lines represent equal loss values.

MoEs.

# 7. Related Work

## 7.1. Scaling Laws for Language Models

Scaling laws have proven to be a powerful framework for understanding and predicting the performance of language models. Existing studies, such as Kaplan et al. (2020) and Hoffmann et al. (2022), reveal that power-law relationships govern model performance as a function of factors like model size, data size, and compute budget, offering predictable performance improvements with increased resources.

Hoffmann et al. (2022) emphasizes the critical balance between model size and the number of training tokens when the training compute budget is fixed, showing that scaling the model without corresponding data increases can lead to suboptimal performance. Additionally, DeepSeek-AI (2024) explores more nuanced scaling behaviors by incorporating data quality, demonstrating that higher-quality data allows for more efficient scaling, and thus, a larger portion of the compute budget should be allocated to increasing model size.

Recent work extends scaling law analysis to specialized contexts, including over-training (Gadre et al., 2024), downstream task performance, and multilingual or multi-modal settings, where scaling laws provide valuable insights and can be adapted to address specific challenges.

## 7.2. Scaling Laws for MoEs

Mixture-of-Experts (MoE) models (Shazeer et al., 2017; Lepikhin et al., 2021; Fedus et al., 2022; DeepSeek-AI, 2025) have emerged as a powerful architecture for language modeling, primarily because they decouple computational cost from parameter count. This separation between parameters and FLOPs per token in MoE architectures calls for scaling laws that can accurately factor in the contributions of both.

Previous research on the scaling behavior of MoE models has established foundational scaling laws, incorporating factors such as total parameter count, the number of experts, and the granularity of these experts (Clark et al., 2022; Ludziejewski et al., 2024; Wang et al., 2024). However, these studies typically assume a fixed configuration for other critical variables influencing FLOPs per token, such as the number of active experts per input. In contrast, we propose a generalized scaling law that considers variables like active parameter count and sparsity level, thereby expanding the applicability of MoE scaling laws.

A common theme in the literature suggests that training sparser models—achieved by increasing the number of smaller experts—offers significant gains in efficiency for both pretraining and inference phases. Through a comprehensive large-scale study, we provide empirical evidence for this, analyzing the impact of sparsity level on efficiency and defining optimal configurations.

Supporting this, Du et al. (2021) demonstrates GLaM's superior efficiency and performance compared to GPT-3, showing that MoE architectures can achieve high performance with significantly lower computational and energy

costs. Further insights are offered by Clark et al. (2022), who analyze scaling behaviors across various MoE routing techniques. While their study finds that MoEs generally outperform dense models, it also notes diminishing benefits as base model sizes grow. Ludziejewski et al. (2024) challenge this conclusion, attributing the diminished returns partly to the fixed number of training tokens across models and constant expert sizes. By introducing "granularity" and adjusting training durations, they demonstrate that MoEs can outperform dense models across any compute budget, debunking the notion of diminishing returns for MoEs with adaptive expert configurations. More recently, Jelassi et al. (2024) finds that, on downstream tasks, MoEs scale efficiently with the number of experts (i.e., increasing sparsity) on memorization tasks, but their reasoning capabilities saturate and lag behind dense models on tasks requiring complex reasoning when compared based on total number of parameters.

Another approach by He (2024) explores the benefits of training MoEs with larger numbers of smaller experts rather than the conventional setup of fewer, larger experts. They introduce Parameter Efficient Expert Retrieval (PEER), a novel routing mechanism designed to tackle the computational and optimization challenges that arise when handling a high number of experts, thus enabling efficient scaling of MoE models.

Lastly, Yun et al. (2024) draws attention to the increased inference costs associated with scaling MoEs by adding experts. While additional experts may not substantially affect training costs, they can inflate inference costs, thereby diminishing deployment efficiency. To address this, the study proposes an over-trained budget allocation strategy, optimizing MoE models for both performance and efficiency in deployment.

## 8. Conclusion

In this paper, we investigated the optimal trade-off between parameters and compute per example for maximizing model capacity. Our findings indicate that sparsity, as a knob that controls FLOPs per example in MoEs, is a powerful mechanism for optimizing model performance under constrained training compute budgets. By balancing the total number of parameters, compute, and sparsity, MoEs can be scaled more effectively. These insights provide valuable guidance for scaling language models, especially for MoEs, where the trade-offs between parameters and FLOPs must be carefully managed.

MoEs were originally introduced to allow increasing model capacity without a significant increase in inference cost. Our experiments show that under fixed total training compute budget increasing sparsity in MoEs leads to smaller FLOPs

per example, higher number of parameters, and lower pre-training loss simultaneously. In other words, in the context of MoEs, if there are no constraints on the total number of parameters, increasing the capacity of the model through parameter count seem to be the optimal strategy if lower pretraining loss is the main goal. On the other hand, when comparing how well the pretraining performance transfers to various downstream tasks, denser models exhibit better transfer performance on certain types of task that potentially rely on deeper processing of the input vs the knowledge stored in the parameters of the model. This potentially signals the importance of the role of FLOPs per example in increasing the capacity of the model during inference. Our experiments demonstrate that MoEs use Chain-of-Thought prompting more effectively than dense models, achieving better performance when allocated additional computational resources during inference. This observation reveals an interesting direction to improve the performance efficiency of MoEs at inference time.

Future work will focus on determining the optimal balance between FLOPs per example and parameter count, with an emphasis on conducting in-depth analyses of model performance across diverse downstream tasks. A key direction will involve exploring strategies to balance parameter allocation and computational demands to minimize inference costs. Developing scaling law studies to identify optimal approaches for achieving efficiency and performance during inference represents a critical area for further investigation.

Another important avenue will be to examine how the findings on the role of sparsity in MoEs generalize to architectures or approaches that employ different mechanisms for independently adjusting FLOPs per example and the number of trainable parameters. Additionally, an intriguing direction for future exploration is the study of scaling behaviors in models that enable negative sparsity values through parameter sharing.

## Impact Statement

This paper presents work whose goal is to advance the field of Machine Learning. There are many potential societal consequences of our work, none which we feel must be specifically highlighted here.

## Acknowledgments

The authors would like to thank Vaishaal Shankar, Fartash Faghri, Skyler Seto, Mustafa Shukor, Amitis Shidani, David Grangier, Etai Littwin, Alexander Toshev and Preetum Nakkiran for their insightful discussions, feedback and technical support that significantly contributed to the development of this paper.

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

# A. Preliminaries

## A.1. Notation and Terminology

To aid readability, we provide a list of key symbols used throughout this paper.

| Symbol | Description |
|---|---|
| $N$ | Total number of model parameters |
| $N_a$ | Active number of model parameters |
| $S$ | Sparsity level (ratio of non-active to total experts) |
| $S^*$ | Optimal sparsity level |
| $L$ | Pretraining Loss (Categorical Cross-Entropy) |
| $L^*$ | Optimal pretraining loss |
| $C$ | Total training compute budget (in FLOPs) |
| $N^*$ | Optimal total number of parameters |
| $N_a^*$ | Optimal active number of parameters |
| $E$ | Expansion factor (number of experts per MoE layer) |
| $K$ | Number of selected experts per token |
| $G$ | Granularity of experts (size relative to base MLP) |
| $D$ | Dataset size (number of training tokens) |
| $\alpha, \beta, \gamma, \lambda, \delta, a, b, c, d, e$ | Coefficients in the parametric scaling law equation |

In this paper, we use the term "compute" in a general sense to refer to computational cost. Unless otherwise specified, "compute" and "FLOPs" (Floating Point Operations) are used interchangeably to quantify this cost.

## A.2. Mixture-of-Expert (MoE) Transformers

Mixture-of-Experts Transformers modify the standard transformer architecture by introducing in the MLP layer. In this design, the experts are MLP (Multi-Layer Perceptron) modules that follow the attention mechanism and are selectively activated for each token. A gating mechanism determines which MLP experts are most relevant for each token, ensuring that only a subset of experts (top-k) is active at any given time, while the rest remain inactive. Below, we provide the notations used throughout the paper for various terms related to training MoEs.

**Total and Active Parameters:** In MoEs, we distinguish between total and active parameters, denoted by $N$ and $N_a$, respectively. The total parameter count, $N$, includes all parameters of the network, encompassing both the experts and the rest of the architecture. The active parameter count, $N_a$, refers to the parameters associated with the active portion of the experts, along with the rest of the network that is always utilized.

**Top-k Expert Selection:** In MoEs, the gating mechanism assigns tokens to a subset of experts using a top-k selection process, where $k$ denotes the number of experts activated for each token. The gate computes a relevance score for each expert, and the top $k$ experts with the highest scores are selected and activated. This selective activation limits the computational overhead by ensuring that only a fraction of the experts are used per token.

**Expansion Factor and Granularity:** The expansion factor, typically denoted by $E$, represents the increase in model capacity due to the inclusion of multiple experts, measured as a multiplicative factor relative to the base dense model. The granularity, $G$, determines the size of each expert relative to the size of the MLP module in the base dense model. The total number of experts in the model is given by $E \times G$, where $E$ scales the capacity and $G$ controls the level of granularity.

**Sparsity ($S$):** In general, sparsity is defined as the ratio of inactive to total parameters. However, in the context of MoEs, we focus on the sparsity of the MLP modules specifically. Therefore, we define the sparsity level as the ratio of inactive to total experts, given by:

$$S = \frac{\text{number of non-active experts}}{\text{number of total experts}}. \tag{7}$$

This definition provides an interpretable measure of sparsity but cannot be directly used to calculate the active parameter count $N_a$ due to the contribution of other parameters in the model that remain unsparsified.

# B. Experimental Setup

We train and evaluate auto-regressive sparse Mixture-of-Experts (MoE) language models of varying sizes and configurations on subsets of the RedPajamaV1 dataset (Together Computer, 2023). The key variables we explore in our experiments are total model parameters $N$, training compute budget $C$, and the MoE sparsity $S$.

**Pre-training data.**    Our models are pre-trained on subsets of the RedPajamaV1 dataset[4] (Together Computer, 2023), which attempts to replicate the LLaMA pre-training data recipe and comprises 1.2 trillion tokens from sources such as Common Crawl, C4, GitHub, and Wikipedia. In all our experiments, the effective dataset size is adjusted based on the training compute budget $C$ and the model size $N$. We tokenize the data using the GPT-NeoX tokenizer (Black et al., 2022), which has a vocabulary size of $50,432$ tokens.

**Model and tokenizer.**    We use auto-regressive transformer-based MoE language models in order to study compute-parameter trade-offs by varying MoE sparsity. We use the Megablocks library (Gale et al., 2023) to train dropless MoEs in which the routing mechanism ensures that all tokens are efficiently routed without being dropped due to routing capacity constraints.

**Optimizer and scheduler.**    We optimize our models using the scale-free Adam optimizer[5] with variable learning rate, a weight decay of $1 \times 10^{-5}$, and fixed Adam-specific parameters $\beta = (0.9, 0.95)$ and $\varepsilon = 1 \times 10^{-8}$. We use a learning rate scheduler consisting of a linear warm-up phase followed by a cosine decay. The warm-up phase increases the learning rate from 0 to the base learning rate over a fraction of the total training steps (selected from $\{0.1, 0.05, 0.02\}$). After warm-up, the learning rate decays following a cosine schedule for the remaining training steps.

**Fitting IsoFLOP surfaces.**    Recall that in Section 2, we fit isoFLOP surfaces to predict pretraining loss $L$ as a polynomial function of model size $N$ and MoE sparsity $S$ for a fixed training budget $C$. The polynomial function takes the form

$$L(N, S) = \sum_{i=1}^{\alpha_1} a_i \hat{N}^i + \sum_{i=1}^{\alpha_2} b_i \hat{S}^i + \sum_{i=1}^{\alpha_3} c_i (\hat{N} \cdot \hat{S})^i + d \tag{8}$$

where $\hat{N} = \log N$ and $\hat{S} = -\log(1 - S)$—we find that applying log transformations improves the fit of the resulting IsoFLOP surface. Through a grid search over the polynomial coefficients $\alpha_1, \alpha_2, \alpha_3 \in \{0, 1, 2, 3, 4\}$, we found that the best fit was obtained for $\alpha = \beta = \gamma = 2$, i.e., a quadratic polynomial over $\hat{N}$ and $\hat{S}$. We evaluate the fitted IsoFLOP surfaces in Figure 1 by (a) re-running the fitting procedure $k = 100$ times on randomly subsampled data and (b) evaluating the Pearson correlation between the true and predicted pretraining loss values on a set of held-out data points.

**Hyperparameters.**    We followed established best practices to train MoEs that included carefully searching over important hyperparameters like learning rate, weight decay, warm up schedule. Furthermore, we used a load balancing loss, router-Z loss to stabilize training and QK-normalization to stabilize training. We fix a subset of hyperparameters for which changing values in preliminary experiments (a) did not significantly improve pre-training loss, (b) the optimal value remained the same across several model configurations, or (c) in order to reduce the search space (i.e., limited compute resources). Specifically, we first opted to use $z$-router loss (Zoph et al., 2022) and $qk$-normalization (Wortsman et al., 2023) in order to stabilize training for large MoEs. Second, we fixed MoE router jitter noise to 0, as it did not improve performance. We also fixed our batch size to 1024 for all model sizes.

We swept over hyperparameters that, when adjusted, (a) significantly improved pre-training loss and (b) the optimal values varied across different model configurations. We increase the MoE sparsity by decreasing the number of active experts and/or increasing the number of total experts. We also varied the MoE granularity (Ludziejewski et al., 2024), MoE load balancing regularizer, Adam learning rate, and linear warm-up steps (fraction) in order to improve pre-training loss. The table below summarizes our hyperparameter sweeps:

---

[4]GitHub repository: `https://github.com/togethercomputer/RedPajama-Data`
[5]Scale-free Adam: `https://fabian-sp.github.io/posts/2024/02/decoupling/`

*Table 1.* Hyperparameter configurations and search spaces

| Hyperparameter | Configuration | Search Space |
|---|---|---|
| Sparsity Level | Tuned | $\{0, 25, 50, 75, 90, 95, 98\}\%$ |
| Number of Total Experts | Tuned | Adjusted depending on sparsity |
| Number of Active Experts | Tuned | Adjusted depending on sparsity |
| Granularity | Tuned | $\{1, 2\}$ |
| Learning Rate | Tuned | $[0.003, 0.002, 0.001]$ |
| Load Balancing Factor | Tuned | $\{0.02, 0.05\}$ |
| Warm-up Steps | Tuned | $\{2, 5, 10\}\%$ |
| Batch Size | Constant | 1024 |
| Jitter Noise | Constant | 0 |
| z-Loss | Constant | 0 |
| z-Router Loss | Constant | 0.001 |
| QK Norm | Constant | Applied |

It is also noteworthy that, in this paper, we have prioritized training compute-optimal models, in contrast to many published results on large language models (LLMs), which often rely on over-trained models. As a result, the performance of the models we use for the analysis in this paper is not directly comparable to those of other studies, where they overtrain smaller language models, to reduce the cost of inference relative to training.

## C. Estimating Mixture-of-Expert (MoE) FLOPs

Similar to prior work on scaling laws (e.g., Kaplan et al. (2020); Hoffmann et al. (2022); Ludziejewski et al. (2024)), we use theoretical FLOP estimates as proxies for training and inference costs of language models. In this section, we (a) outline our methodology for estimating FLOPs for MoEs and (b) show that the proposed estimator closely approximates empirical FLOPs of large-scale MoEs.

**Setup and notation.** Consider an MoE model with $n_{\text{layers}}$ MoE layers, each with an embedding dimension of $d_{\text{model}}$. We denote the number of total experts and active experts in each MoE layer by $E_{\text{total}}$ and $E_{\text{active}}$ respectively. Following Ludziejewski et al. (2024), we let $G$ denote the MoE granularity, which defaults to 1 and controls the size of each expert relative to the size of a feed-forward layer in an equivalent dense transformer. In order to change sparsity in a more granular manner, we treat the number of active experts as an independent variable that does not scale with granularity $G$. In our experiments, we use a vocabulary size $n_{\text{vocab}} = 50,432$, a context length $n_{\text{ctx}}$ of 2048, and GLU modules (Gated Linear Units) (Shazeer et al., 2017) over feed-forward modules as the architecture of choice for MoE experts. We also set the (a) hidden dimension of each GLU expert $d_{\text{ffn}}$ to $4 \cdot d_{\text{model}}$ and (b) instantiate MoEs where the number of attention heads $n_{\text{heads}}$ times the dimensionality for each head $d_{\text{head}}$ equals $d_{\text{model}}$, i.e., $n_{\text{heads}} d_{\text{head}} = d_{\text{model}}$.

**Estimating module-specific FLOPs.** To estimate the FLOPs of a given MoE model, we first individually estimate the FLOPs per token incurred by a forward *and* backward pass through every module in MoEs. Then, we aggregate these estimates to obtain the final estimator for the FLOPs per token incurred by a forward *and* backward pass through the model.

Like in prior work on scaling laws (Kaplan et al., 2020; Hoffmann et al., 2022), we take a two-step approach to estimate module-specific FLOPs. Given a module, we first estimate the number of parameters in the module and then scale this with an appropriate constant corresponding to the number of add-multiply operations per parameter through a forward and backward pass of the given module. We also omit non-leading terms such as non-linearities, biases, and layer normalization in our estimation. We estimate the FLOPs per token for attention modules, MoE routers, MoE experts, and the final un-embedding layer as follows:

1. **Attention module.** We estimate the FLOPs incurred via the QKV (and final) projections, attention logits, and attention values of all heads in a multi-head attention module as follows.

   - *QKV (and final) projections.* These projections involve $4 \cdot d_{\text{model}} n_{\text{heads}} d_{\text{heads}} = 4 d_{\text{model}}^2$ parameters. Following Kaplan et al. (2020), we use the multiplicative constant $C = 6$ to account for the add-multiply operations per parameter in a forward and backward pass through linear modules, resulting in a FLOPs-per-token estimate of $4 \cdot C \cdot d_{\text{model}}^2$.
   - *Attention logits.* The FLOPs required to compute the attention logits for all $n_{\text{ctx}}$ tokens equals $C \cdot n_{\text{ctx}}^2 d_{\text{model}}$ FLOPs, making the FLOP-per-token estimate equal to $C \cdot n_{\text{ctx}} d_{\text{model}}$.
   - *Attention values.* The computation of attention values requires a per-token weighted sum over $n_{\text{ctx}}$ $d_{\text{model}}$-dimensional vectors, making the estimate $C \cdot n_{\text{ctx}} d_{\text{model}}$.

2. **MoE module.** Given an MoE layer, we estimate the FLOPs incurred by its router and all experts separately.

   - *Router.* The MoE routing linearly maps a $d_{\text{model}}$-dimensional token embedding to a $E_{\text{total}}$-dimensional logit vector, which is subsequently used to map the token to $E_{\text{active}}$ active experts. Following Ludziejewski et al. (2024), we use a multiplicative constant $R = 14$ that accounts for the add-multiply-route operations per router parameter. The resulting FLOP estimate equals $R \cdot d_{\text{model}} E_{\text{total}}$
   - *Experts.* Each MoE experts corresponds to a GLU module (Shazeer et al., 2017) with $d_{\text{ffn}} = 4 \cdot d_{\text{model}}$. Since there are $E_{\text{active}}$ active experts with granularity $G$, each involving three linear projections, this results in a FLOP estimate of $1/G \cdot 3 \cdot E_{\text{active}} \cdot C \cdot d_{\text{model}} d_{\text{ffn}} = 12C/G \cdot E_{\text{active}} \cdot d_{\text{model}}^2$.

3. **Un-embedding layer.** The un-embedding linear layer maps the final $d_{\text{model}}$-dimensional embedding of a token to $n_{\text{vocab}}$-dimensional logits, making the FLOPs-per-token $C \cdot n_{\text{vocab}} d_{\text{model}}$.

**Estimating MoE FLOPs.** We can aggregate the module-level FLOP estimates described above to estimate the FLOPs per token required for a single forward and backward pass through a given MoE model as follows:

$$n_{\text{layer}}\left(4C d_{\text{model}}^2 + 2C d_{\text{model}} n_{\text{ctx}} + 12C/G E_{\text{active}} d_{\text{model}}^2 + R d_{\text{model}} E_{\text{total}}\right) + C n_{\text{vocab}} d_{\text{model}}$$

When $E_{\text{total}}/d_{\text{model}}$ is small, which is typically the case in practice, the FLOPs induced by MoE routing can be ignored as they contribute negligibly to the estimator. This allows us to simplify the estimator to:

$$\text{MoE FLOPs per token} := C \cdot n_{\text{layers}} d_{\text{model}}^2 \left( 4 + \frac{2n_{\text{ctx}}}{d_{\text{model}}} + \frac{12E_{\text{active}}}{G} + \frac{n_{\text{vocab}}}{d_{\text{model}} n_{\text{layers}}} \right) \tag{9}$$

**Evaluating $6N_aD$ as a FLOPs-per-token estimator in MoE Models**  For standard dense transformers, the FLOPs are often estimated as $6ND$ (Kaplan et al., 2020; Hoffmann et al., 2022). Given that $D$ is fixed and not adjusted dynamically, $N$ can serve as a relative estimator of FLOPs per token for dense transformer models.

To adapt the $6ND$ estimator for MoE models, we replace $N$ with $N_a$ (the active number of parameters)—the number of parameters used in every forward and backward pass. In Figure 7, we evaluate the accuracy of the $6N_aD$ estimator by plotting the ratio between the MoE FLOPs estimator described in Equation 9 and $6N_aD$ as a function of model size $N$ and a fixed context length $D = 2048$. The results show that, across all sparsity levels, the ratio remains close to one, and the gap between the two estimators decreases as model size $N$ increases.

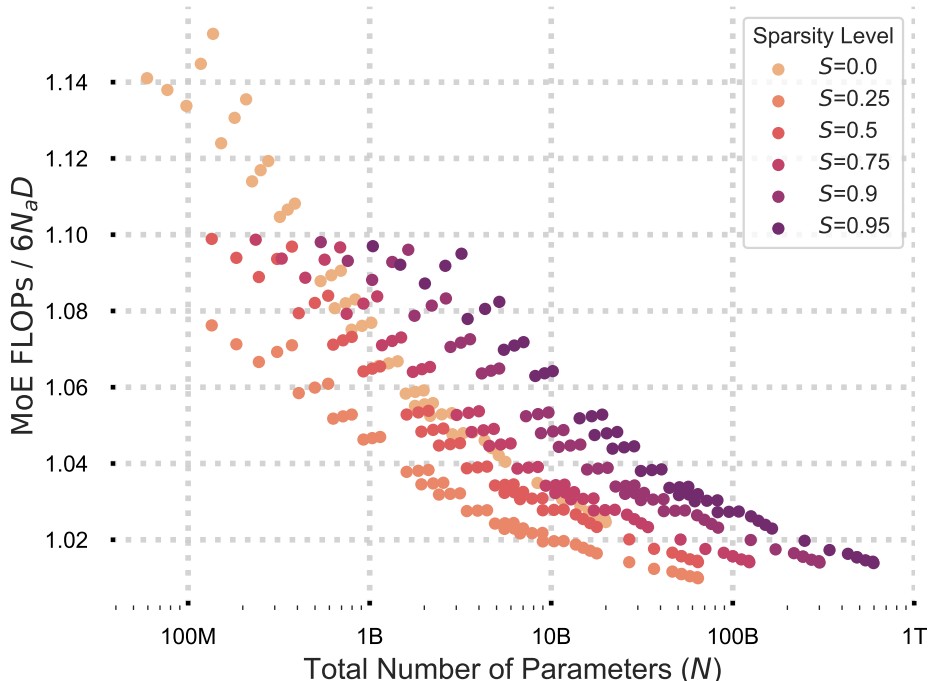

*Figure 7.* **Accuracy of $6N_aD$ FLOPs Estimator for MoEs**. Ratio of the MoE FLOPs estimator (Equation 9) to the $6N_aD$ estimator as a function of the total number of parameters, for a fixed context length of $D = 2048$, used in our experiments.

# D. Additional Analysis

### D.1. Interplay between parameters and FLOPs per example

Recall that in Section 2, we showed that isoFLOP curves were predictive of pretraining loss for different parameter counts and sparsity levels. In this section, we show similar results with additional training compute budgets.

1. In Figure 8, we first show that IsoFLOP surfaces mapping model size $N$ and sparsity level $S$ to pre-training loss $L$ are predictive in a similar way for all training compute budgets that we consider, ranging from 3e19 to 1e21 FLOPs.

2. In Figure 9, we analyze the fitted IsoFLOP surfaces (one for each training budget) and find that the (a) effect of model size $N$ on optimal MoE sparsity $S^*$ and (b) the effect of MoE sparsity $S$ on the optimal total and active parameters, $N*$ and $N_a^*$, is similar for all training budgets.

### D.2. Effect of training budget and model size on optimal MoE sparsity

Recall that in Section 3, we demonstrated how the relationship between optimal total parameters $N*$, optimal active parameters $N*_a$, and optimal pretraining loss $L$ predictably changes as a function of sparsity $S$ and training budget $C$. In this section, we use the fitted isoFLOP surfaces to analyze how the optimal MoE sparsity $S^*$ changes as a function of total parameters $N$ and training budget $C$, as shown in Figure 4. Our main findings are:

- Across all training budgets (ranging from 3e19 to 1e21 FLOPs), increasing the total parameters $N$ leads to an increase in the optimal sparsity level $S^*$.

- For a fixed model size (i.e., total parameters $N$), increasing the training budget $C$ generally reduces the optimal sparsity level $S^*$.

- The relationship between model size $N$ and optimal $S*$ is not linear. For smaller models (up to about $500 \cdot 10^6$ parameters), the optimal sparsity remains at 0 (i.e., dense) for most compute budgets.

### D.3. Effect of sparsity on downstream task performance

In Section 4, we analyzed the relationship between upstream pre-training loss and downstream task performance across different MoE sparsity levels. We found that language understanding and world knowledge tasks generally showed a strong correlation between upstream and downstream performance, while reading comprehension tasks seemed to favor denser models to some extent.

In this section, we provide additional plots for a broader range of tasks within each category to further support our findings. We consider the following tasks:

- **Common Sense Reasoning**: PIQA, CommonSenseQA, OpenBookQA, COPA
- **Language Understanding**: LAMBADA, HellaSwag, Winograd, Winogrande
- **Reading Comprehension**: SQuAD, CoQA, BoolQ
- **World Knowledge**: TruthfulQA, ARC-Easy, ARC-Challenge

Figure 10 shows the relationship between upstream pre-training loss and downstream task performance for these additional tasks. Each row corresponds to a task category and each subplot represents a different task, with points colored according to MoE sparsity $S$. The $x$-axis represents the upstream pre-training loss, while the $y$-axis shows the downstream task performance metric (usually accuracy or error rate). These results supplement our main findings from Section 4:

- We observe consistent trends across tasks within each category, with language understanding and world knowledge tasks showing strong correlations between upstream and downstream performance regardless of sparsity.

- Reading comprehension tasks continue to show a slight advantage for denser models, while common sense reasoning tasks (which can be considered part of the symbolic problem-solving category) show more varied relationships between upstream and downstream performance.

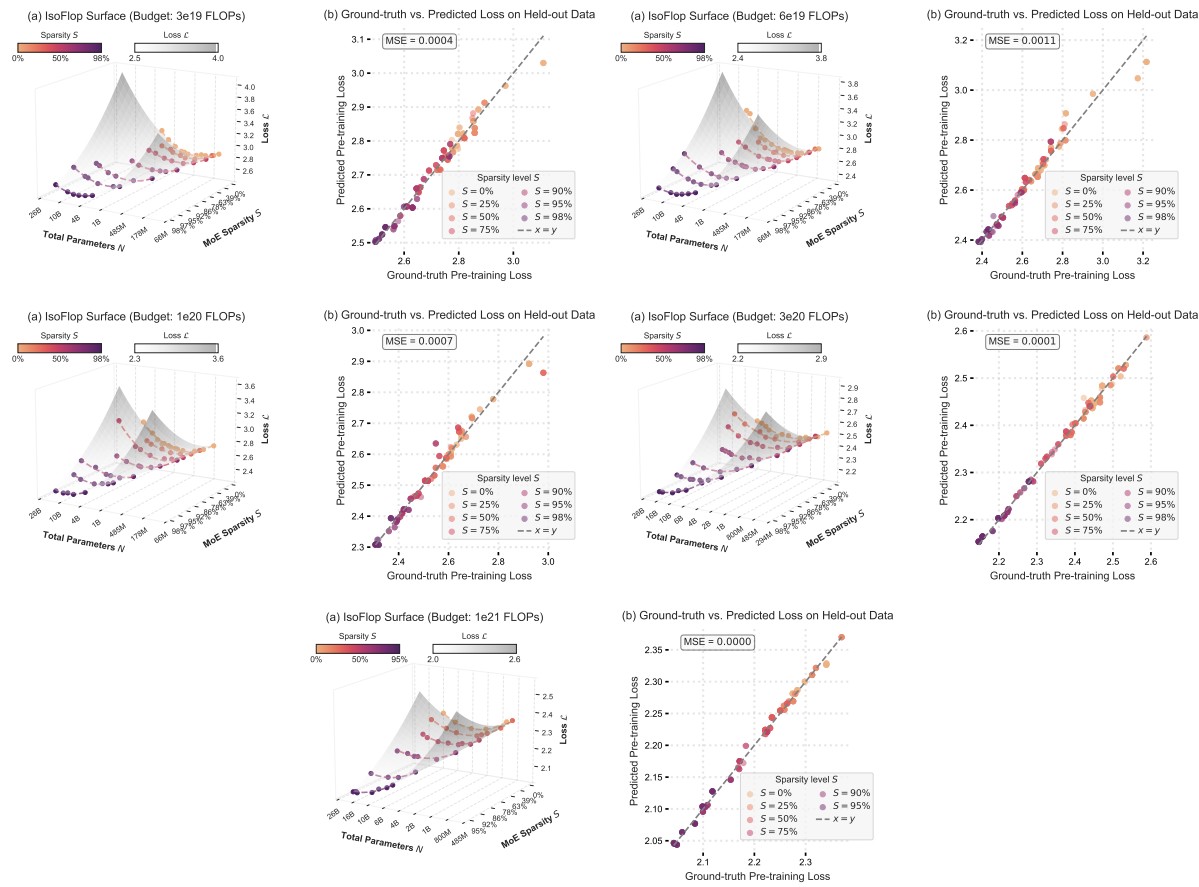

*Figure 8.* **IsoFLOP surfaces over total parameters $N$, MoE sparsity $S$, and pretraining loss $L$ for different compute budgets**. The rows correspond to IsoFLOP surface fitted using models trained with a budget of 3e19, 6e19, 1e20, 3e20, and 1e21. The subplots on the left visualize IsoFLOP surfaces mapping total parameters $N$ and sparsity level $S$ to pretraining loss $L$. The subplots on the right correlate the ground-truth pretraining loss with the estimated pretraining loss on held-out data. Taken together, these results show that isoFLOP surfaces are accurate proxies for understanding how model size and MoE sparsity jointly impact pretraining loss.

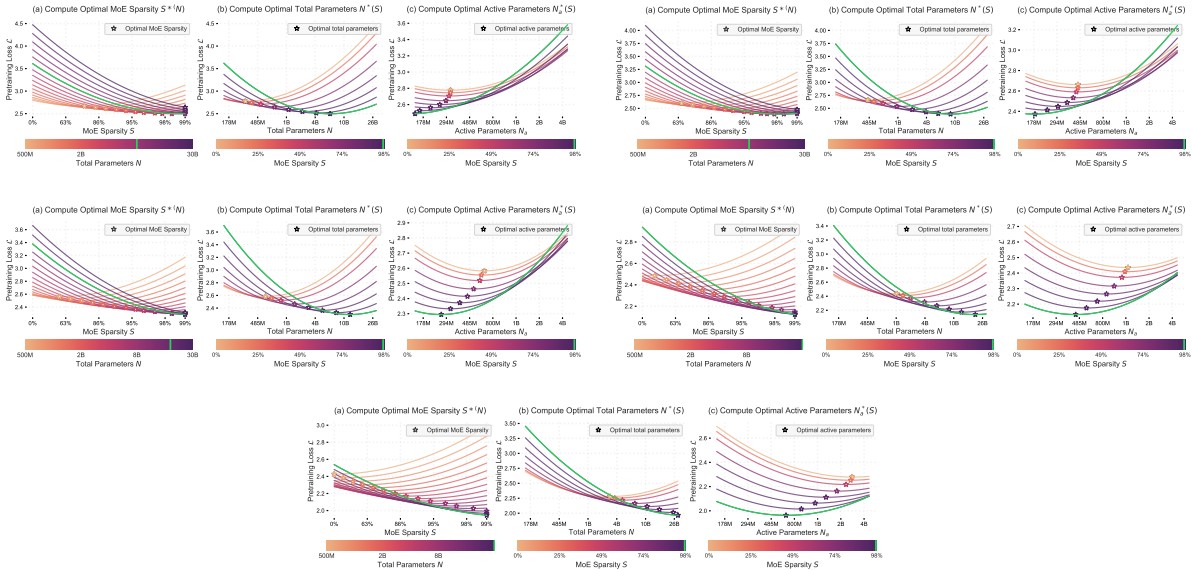

*Figure 9.* **Optimal MoE configurations predictably change with training compute budget.** Each row corresponds to an analysis of how optimal MoE sparsity $S^*$, total parameters $N^*$, and active parameters $N_a^*$ change for a given training budget. The subplots on the left show that (a) increasing the training budget increases the model size $N$ (denoted with black dots) with the minimum pretraining loss and (b) for models smaller than a threshold (which increases with training budget), dense models (i.e., $0\%$ sparsity) fare better than sparse MoEs. The subplots in the second and third panel show that (a) increasing MoE sparsity increases the optimal total parameters $N^*$ and decreases the optimal active parameters $N_a^*$. In both cases, for a fixed sparsity level, increasing the budget shifts increases the optimal total and active parameters.

## D.4. Comparing IsoFLOP Surface Analysis with Independent 2d IsoFLOPs

Recall that in Section 2, we used IsoFLOP surfaces that predict pre-training loss across varying parameter counts and sparsity levels to understand how optimal sparsity and optimal model size depend on each other.

In this section, we evaluate whether these findings remain consistent when we do not rely on fitted IsoFLOP surfaces. Specifically, similar to Approach II in Hoffmann et al. (2022), we directly fit univariate quadratic functions that map model size $N$ to pre-training loss $L$, independently for each sparsity level and training compute budget. We then assess these univariate fits to determine whether our findings in Section 2 hold.

- In Figure 12, each row shows how the optimal total and active parameters change as a function of MoE sparsity for fixed training budgets. As in our findings from Section 2 (Figure 2), increasing sparsity increases the optimal total parameters while decreasing the optimal active parameters. Moreover, larger compute budgets still result in higher optimal total and active parameters, regardless of the sparsity level.

- Furthermore, in Figure 11, we observe that across all training compute budgets, increasing sparsity reduces the optimal pre-training loss. This is consistent with the trends identified in Section 3 (Figure 3), thereby validating our earlier results.

## E. Does Chain-of-Thought prompting benefit sparse MoEs more than dense models?

In Section 4, we observed that dense models fare marginally better than sparse MoEs on reading comprehension tasks, potentially due to the higher inference-time compute of a dense model than a perplexity-matched sparse MoE. Then, in Section 6, we hypothesized that alternative strategies to increase inference-time compute may reduce the gap between sparse MoEs and dense models on such tasks. In this section, we test this hypothesis by leveraging a "length-controlled" variant of few-shot Chain-of-Thought (CoT) prompting to indirectly control inference-time compute. We then use this to study the effect of inference-time compute on downstream task performance of dense and MoE models.

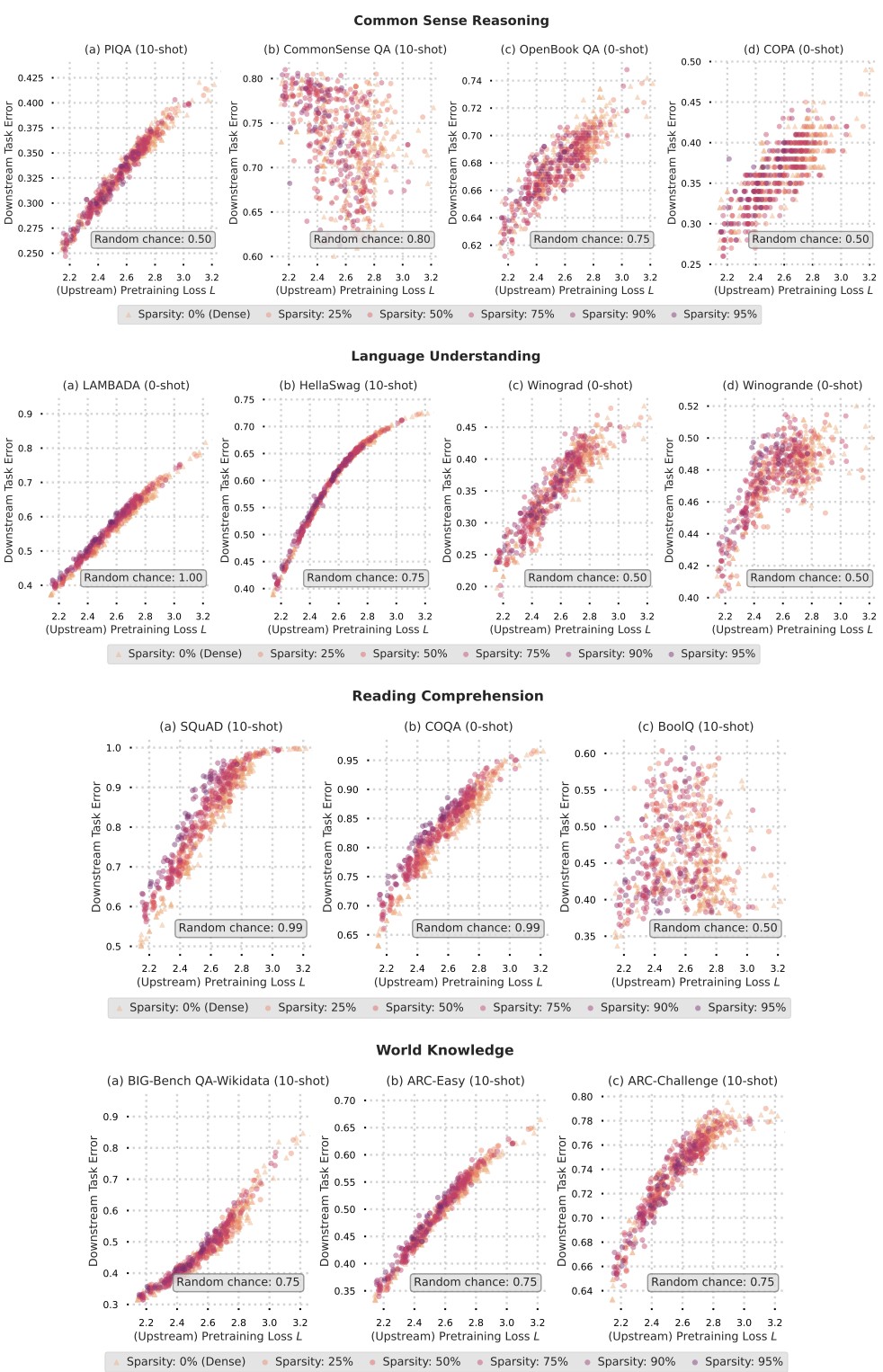

*Figure 10.* **Downstream task performance vs. upstream pre-training loss.** Each subplot shows the relationship between upstream pre-training loss (x-axis) and downstream task performance (y-axis) for a specific task. Similar to our results in Section 4, we find that the MoE sparsity level does not change the relationship between upstream pre-training loss and downstream task performance.

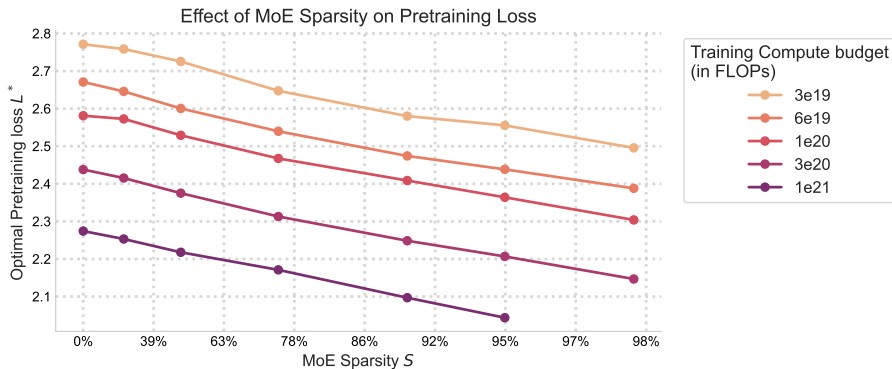

*Figure 11.* **Effect of MoE sparsity on pretraining loss across different training compute budgets**. As sparsity increases, the validation loss decreases for all compute budgets, with larger budgets (darker lines) achieving lower losses at each sparsity level. This trend is consistent with the findings from Section 3, demonstrating that increasing sparsity reduces the optimal pretraining loss across all compute budgets.

**Experiment setup.** We evaluate Qwen1.5 models (Bai et al., 2023) on the GSM8k dataset (Cobbe et al., 2021) to study the effect of few-shot CoT prompting (Wei et al., 2022b) on downstream task performance. We look at the effect of increasing inference-time compute (via CoT prompting) on GSM8k performance of dense models with sizes ranging from 0.5B to 14B and a 5x2.7B sparse MoE. We also use 10 fixed examples from the GSM8k train split as few-shot examples for all runs.

**Length-controlled CoT prompting enables control over inference-time compute.** Given an instruction-tuned model and a problem from the GSM8k dataset, we control the inference-time compute of the model by controlling the number of tokens generated to output the final answer to the given problem. To do so, we observe that providing instructions via system prompts (with few-shot CoT prompting) does not effectively control the number of generated tokens and, as a result, inference-time compute. We also observe that the average number of tokens in the few-shot GSM8k answers (provided in-context) strongly influences the number of tokens generated by the model to solve the given GSM8k question. Therefore, similar to work on designing GSM8k variants to analyze language modeling phenomena (Mirzadeh et al., 2024; Li et al., 2024; Zhang et al., 2024), we prompt an instruction-tuned model—Llama-3.1-70b (Dubey et al., 2024) in our experiment—to rewrite GSM8k answers in approximately $k \in \{5, \ldots, 100\}$ words. Then, we use the paraphrased GSM8k examples as few-shot examples to indirectly control the number of generated tokens. As shown in Figure 13(a), this approach enables systematic control over the length of paraphrased answers. The strong correlation ($\rho = 0.88$) between few-shot and generated answer lengths in Figure 13(b) validates that our approach effectively modulates inference-time compute.

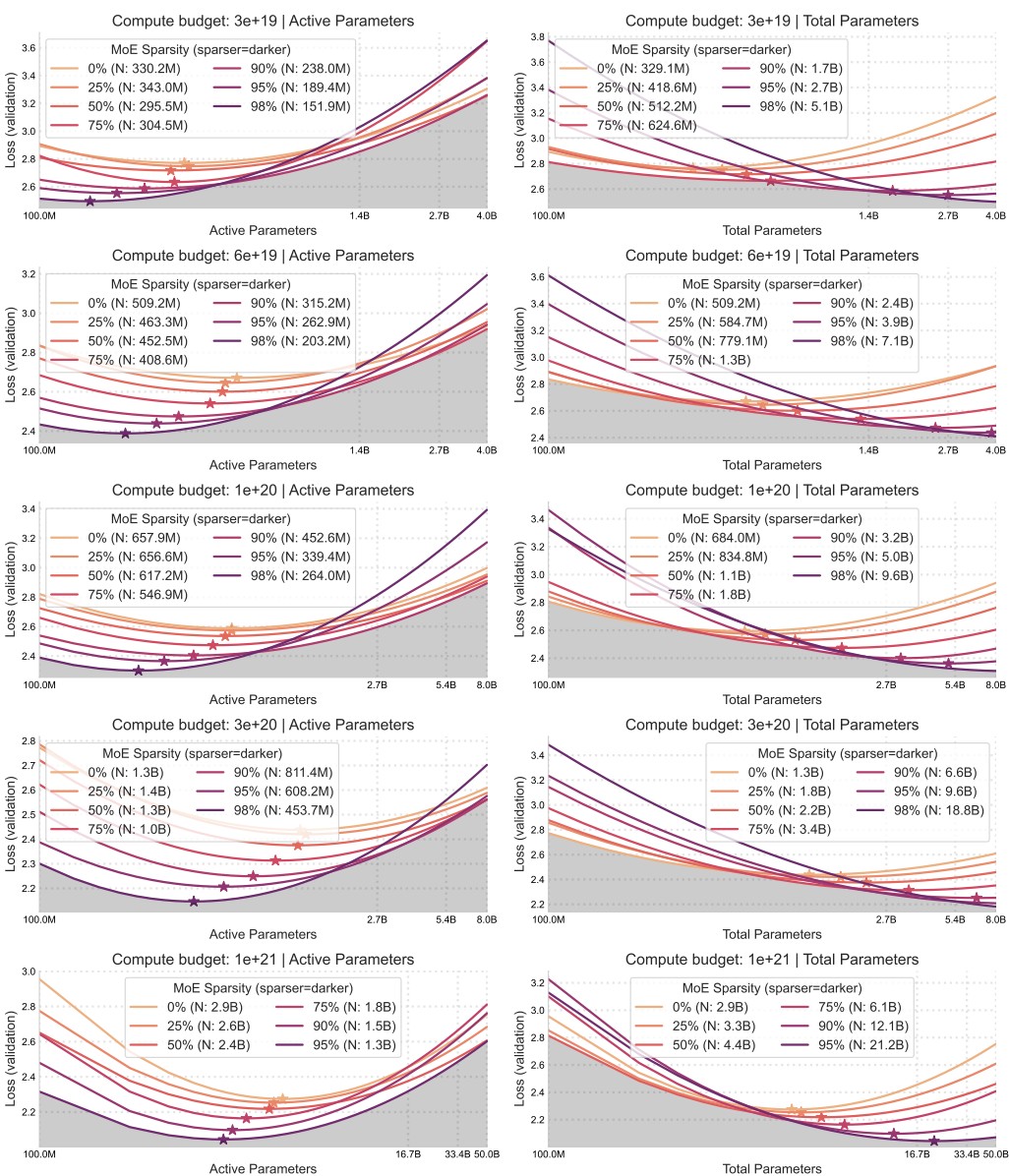

*Figure 12.* **Effect of MoE sparsity on optimal total and active parameters across different training compute budgets.** Each row shows the change in total and active parameters as a function of sparsity level for fixed training budgets. Increasing sparsity leads to an increase in the optimal total parameters while reducing the optimal active parameters, consistent with our findings in Section 2 (Figure 2). Larger training compute budgets result in higher optimal (total and active) parameters across all sparsity levels.

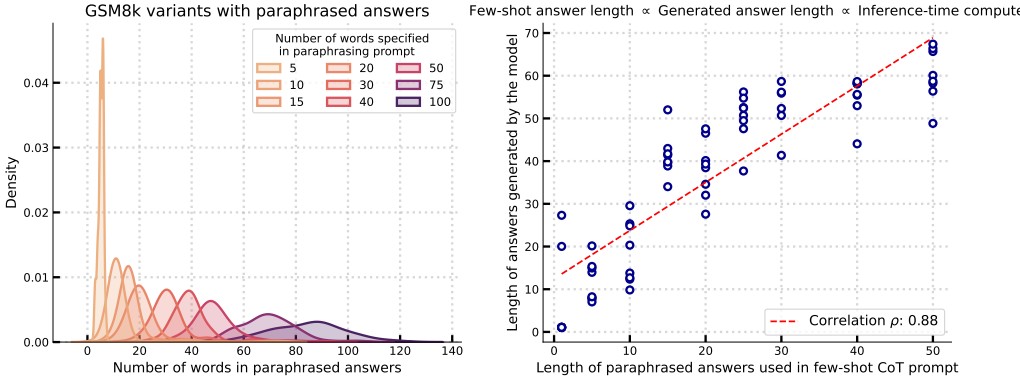

*Figure 13.* **Varying inference-time compute via length-controlled Chain-of-Thought prompting.** We can control inference-time compute (number of generated tokens) via Chain-of-Thought prompting in two steps: (a) generating paraphrased GSM8k answers of varying lengths (5-100 words) and (b) using paraphrased answers as few-shot examples in CoT prompts to influence the length of generated answers determine the model's output length ($\rho = 0.88$ correlation). The systematic shift in answer length distributions in subplot (a) and the linear relationship between few-shot answer length and generated answer lenth ($\rho = 0.88$ correlation) in (b) validate that our prompting approach effectively modulates inference-time compute, enabling controlled studies of its impact on model performance.

**Effect of length-controlled CoT on downstream task performance.** We investigate how inference-time compute affects GSM8k performance across Qwen1.5 models using 10-shot length-controlled CoT prompting, varying target answer lengths $k$ from 1 to roughly 70 words on average. As shown in Figure 14, when we increase inference-time compute indirectly through longer generated answers ($x$-axis), we observe a roughly linear improvement in GSM8k performance ($y$-axis) across all model sizes. The linear fits for each model also indicate a pattern through their slopes $m$: larger dense models benefit more from additional inference-time compute. This effect is particularly striking when comparing models at the extremes (i.e., Qwen1.5-0.5B versus Qwen1.5-14B).

To analyze whether inference-time compute affects dense and sparse MoE models differently, we examine how the "performance" slope $m$ (i.e., accuracy gain per generated word) varies with model size. While the Qwen1.5-5x2.7B MoE cannot be directly compared to dense models due to different active parameter counts, we can account for this by plotting $m$ against the number of active parameters for both architectures. As shown in Figure 15, when controlling for active parameter count, the MoE model exhibits a higher slope than would be expected from interpolating between dense models. This suggests that MoE models benefit more from dynamically increased inference-time compute compared to dense models with equivalent active parameters.

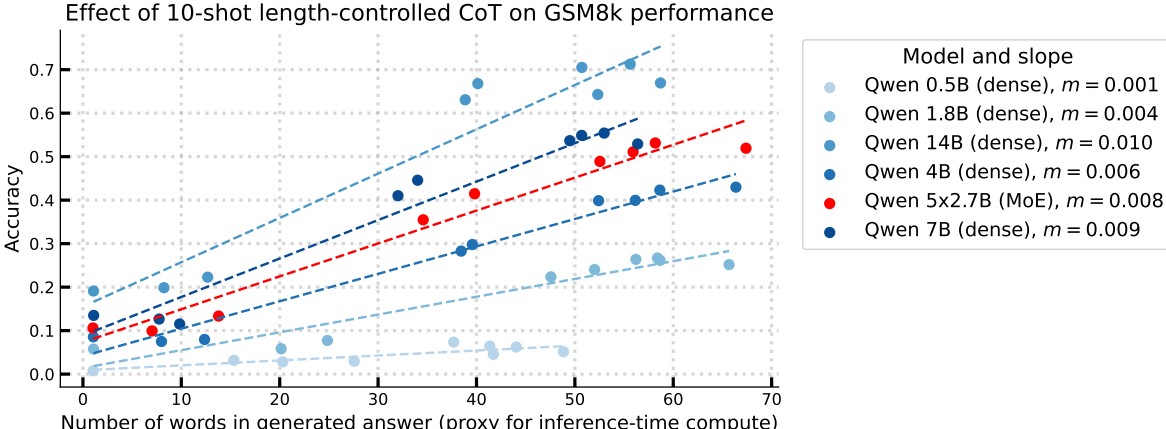

*Figure 14.* **Effect of length-controlled CoT prompting on GSM8k performance across model scales.** We evaluate the relationship between inference-time compute (controlled via answer length) and GSM8k accuracy for dense Qwen1.5 models (0.5B-14B parameters) and a 5x2.7B sparse MoE. For all models, increased inference-time compute improves accuracy roughly linearly, with slopes $m$ indicating the marginal effect. Larger dense models show steeper slopes, demonstrating they benefit more from additional inference-time compute—for example, the 14B model's performance improves from 20% to 70% as answer length increases, while the 0.5B model improves only from 1% to 5%.

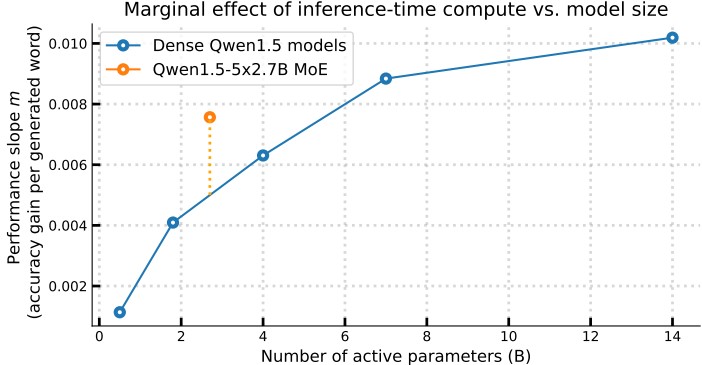

*Figure 15.* **Sparse MoEs benefit more from increased inference-time compute than dense models.** We plot the marginal effect of inference-time compute (accuracy gain per generated word) against model size for dense Qwen1.5 models and a 5x2.7B sparse Qwen1.5 MoE. When controlling for the number of active parameters, the MoE model (orange) shows a higher performance slope than would be expected from interpolating between dense models (blue), suggesting that sparse MoEs benefit more from dynamically increased inference-time compute via strategies like CoT prompting.

# F. Incorporating Sparsity into Scaling Laws

Table 2 shows the parameters used to initialize L-BFGS used to fit the proposed parametric scaling law given in Equation 6. Table 3 shows the estimated parameters for the parameteric model. We use a held out dataset that consists of data points for models with sparsity value $S = 0.98$ to validate the performance of the estimated model coefficients. The mean squared error and the Huber loss error on the dataset used to fit the model is 0.00056 and 0.0036 respectively and 0.0058 and 0.0011 respectively on the out-of-sample validation set. The quality of the fit measured via the $R^2$ metric is 99% on fitting data and 68% on the held out validation dataset.

*Table 2.* Initial values used to estimate coefficients in Equation 6.

| Coefficients | Initial Values |
|---|---|
| $\log(a), \log(b), \log(c), \log(d)$ | $[0, 10, 20]$ |
| $\alpha, \beta, \gamma$ | $[0, 0.25, 0.5, 0.75, 1, 1.25]$ |
| $\lambda, \delta$ | $[-1, -0.5, 0, 0.5, 1]$ |
| $\log(e)$ | 1.5 |

*Table 3.* Estimated values for coefficients in Equation 6.

| Coefficient | Estimate |
|---|---|
| $\alpha$ | 0.5962 |
| $\beta$ | 0.3954 |
| $\lambda$ | -0.1666 |
| $\delta$ | 0.1603 |
| $\gamma$ | 0.1595 |
| a | 16612.50 |
| b | 5455.67 |
| c | 0.4598 |
| d | 17.26 |
| e | 0.94 |

