# OpenReview forum: "Parameters vs FLOPs: Scaling Laws for Optimal Sparsity for Mixture-of-Experts Language Models"
_ICML.cc/2025/Conference — ICML 2025 poster_

### Official Review · Reviewer_TF5J · 2025-02-28

**Overall Recommendation:** 3

**Summary:**

The paper investigates the optimal trade-off between the total number of parameters and FLOPs per example in Mixture-of-Experts (MoE) models under a fixed pretraining FLOPs budget. Experimental results show that increasing total model parameters (i.e., increasing sparsity and reducing active parameters per input) leads to lower pretraining loss for a given compute budget. The paper also studies the relevant impact on the downstream tasks.

**Claims And Evidence:**

Yes.

**Essential References Not Discussed:**

I'm not aware of such related works.

**Experimental Designs Or Analyses:**

Experimental designs are sound, but there are a few limitations:

- One limitation is that the experiments focus primarily on theoretical FLOP estimates, without a thorough quantitative analysis of memory and communication overheads. Those factors are critical for MoE models with large numbers of parameters.

- The paper focuses on a specific MoE implementation, which may limit the generalizability of the findings. Comparing different routing mechanisms or expert configurations would provide valuable insights into whether the observed relationships between sparsity, parameters, and performance hold across different MoE variants. That being said, I'm fine with the current paper using one specific MoE implementation as an initial study.

**Methods And Evaluation Criteria:**

Yes.

**Other Comments Or Suggestions:**

## Presentation issues:
- Line 45 (right): missing a space between the reference and "of".
- Figure 1b: I recommend to use $N_a$ to denote the number of active parameters in this figure so that it looks consistent to others.
- Line 81 (right) perform worse o. $\to$ perform worse.
- Line 325 (left): demonstrates  $\to$ demonstrate
- Line 403 (left): supports $\to$ support

**Other Strengths And Weaknesses:**

## Strengths
- Timely empirical study on an important research topic.
- Systematic experiments on a wide range of settings.
- I particularly like the analysis of optimal sparsity for a given model size. In my opinion, it represents one of the most practically valuable contributions, since it takes real-world constraints like memory and communication requirements into account. The paper would be strengthened by expanding this section and highlighting these real-world implementation considerations more prominently.

## Weaknesses
My overall impression is that while the paper presents extensive experimental results, these findings could have benefited from more structured presentation with clearer takeaway messages. To elaborate,

- The stated research goal is to investigate the optimal trade-off between total parameter count and FLOPs per sample, but the scaling law primarily focuses on total parameters and sparsity. Although sparsity and FLOPs per sample are relevant, this **inconsistency** can make the reader confusing.
- The paper's findings suggest a monotonic relationship where increasing sparsity, increasing total parameters, and decreasing active parameters all lead to better performance. This raises questions about **whether this represents a true "trade-off"** as described in the paper's title. The practical implications of this finding could be more thoroughly discussed - is the recommendation to use increasingly large, extremely sparse models? What are the practical limits to this approach?

- The paper does not provide sufficient discussions on relationship between pre-training loss and downstream performance in both the main paper and the appendix. Figure 10 suggests this relationship varies significantly across tasks and sparsity levels. A more thorough analysis of when and why sparse models transfer differently to downstream tasks compared to dense models would enhance the paper's contributions.
- MoE models are known to present unique training challenges compared to dense models. The paper may benefit from including analysis of how the optimal sparsity levels might be influenced by considerations of training robustness and convergence.

I emphasize that the paper has the potential to be a very strong paper given the systematic empirical study. I am looking forward to an improved version of the paper.

**Questions For Authors:**

- Line 137 (left): _"$K$ is the number of selected experts per token"_: To clarify, does the model always use a fixed $K$ for all tokens?

**Relation To Broader Scientific Literature:**

This paper aligns well with important research topics in both scaling law research and MoE model development, targeting an important open question about optimal sparsity in large-scale language models. The contributions are still valuable for understanding the scaling of MoE models.

**Theoretical Claims:**

No proofs in this paper.

---

> ### Author Rebuttal · Authors · 2025-03-29
>
> We thank the reviewer for their time reviewing our paper. We find it encouraging that the reviewer finds our work valuable and relevant as we systematically study optimal model size and FLOPS-per-token in MoEs. We address the concerns below. Please note that we partially quote or paraphrase reviewer's comments due to space constraints:
>
> > One limitation is that the experiments focus primarily on theoretical FLOP estimates, without a thorough quantitative analysis of memory and communication overheads ...
>
> We have discussed the limitations of using theoretical FLOPs estimates in Section 6.1. We agree that it would be very valuable to benchmark different hardware profiles and leave that as future work.
>
> > The paper focuses on a specific MoE implementation, which may limit the generalizability of the findings ...
>
> We chose the most typical setup for MoEs in our work so we opted to use token-based routing. Examining this choice in detail would indeed be very valuable which we leave as future work
>
> > The stated research goal is to investigate the optimal trade-off between total parameter count and FLOPs per sample, but the scaling law primarily focuses on total parameters and sparsity ...
>
> This is a fair point, and deserves an explanation. While our goal is to understand the relationship between model size and FLOPS-per-example, we study this using a surrogate control knob: increasing the sparsity in MoEs decreases the number of active parameters, which in turn decreases the FLOPs-per-example under settings where compute can take advantage of sparsity. We will make this clear in the paper.
>
> > The paper's findings suggest a monotonic relationship where increasing sparsity, increasing total parameters, and decreasing active parameters all lead to better performance. This raises questions about whether this represents a true "trade-off" as described in the paper's title ...
>
> This is an excellent point raised by the reviewer. Our study finds that as we scale MoE model size, we need to scale up the sparsity value as well. This finding does suggest a tradeoff: we can increase model size but need to reduce the number of active parameters (via sparsity) to gain benefit.  We hope that our work encourages practitioners to invest in building infrastructure for training very large scale MoEs and provides some guidelines on what sparsity value and model sizes to use for a given compute budget. However, given the scale used in our study, we acknowledge that there maybe limitations that will be uncovered when we run models at very large scale, i.e., extrapolating to model sizes significantly larger than what we have used in our study, for which we have not been able to verify how scaling behaves due to compute limitations.
>
>
> > The paper does not provide sufficient discussions on relationship between pre-training loss and downstream performance in both the main paper and the appendix. Figure 10 suggests this relationship varies significantly across tasks and sparsity levels. A more thorough analysis ...
>
> This is another excellent point raised by the reviewer. We were able to conduct some downstream analyses the results of which suggest some task types transfer better than others which is an intriguing finding in our paper, and offered an initial hypothesis that increased test-time compute demands of some tasks may conflict with increasing sparsity (which otherwise improves pre-training loss). A thorough investigation as well as studying test-time interventions would be very interesting but is left as future work.
>
>
> > MoE models are known to present unique training challenges ...
>
> This is a good point raised by the reviewer. We will add to the discussion in our paper that special care needs to be placed on MoE training to ensure that sub-optimal results are not due to poor optimization. We followed established best practices to train MoE that included carefully searching over important hyperparameters like learning rate, weight decay, warm up schedule. Furthermore, we used a load balancing loss, router-Z loss to stabilize training and QK-normalization to stabilize training. All of these details are noted in the appendix of our paper. We acknowledge that training a model at scale (size and sparsity) larger than what is shown in the paper is an involved task.
>
>
> > Presentation issues
>
> We will fix these in the next revision of our paper.
>
> > Line 137 (left): ... To clarify, does the model always use a fixed K for all tokens?
> Yes We use dropless token-based routing
>
> > I emphasize that the paper has the potential to be a very strong paper given the systematic empirical study. I am looking forward to an improved version of the paper.
>
> We appreciate the reviewer’s comment. We hope our rebuttal above addresses reviewer concerns, and promise to update the paper to reflect the reviewer’s thoughtful feedback. If this is satisfactory, we ask the reviewer to consider raising their score.

---

> > ### Comment · Reviewer_TF5J · 2025-04-02
> >
> > Thank you for the response and clarifications. I raise my rating to 3. It would be nice if the authors can also include those clarifications in the updated paper.

---

> > > ### Author Response · Authors · 2025-04-02
> > >
> > > We thank the reviewer for their time and are grateful for their support. We commit to clarifying all the questions raised during the review process in the updated/final version of our paper.

---

### Official Review · Reviewer_6Xtz · 2025-03-11

**Overall Recommendation:** 4

**Summary:**

This paper explores parameter-FLOP trade-offs in sparse MoE LLMs. The author finds that:
1. Increasing sparsity during pretraining improves efficiency and performance under a fixed compute budget.
2. More parameters benefit pretraining, while FLOPs are crucial for inference, especially for reasoning tasks.
3. Optimal sparsity increases with size and compute, approaching full sparsity for large models.
4. Downstream performance correlates with pretraining loss, but denser models excel in tasks like reading comprehension.
5. The authors design a new scaling law incorporating MoE sparsity, guiding efficient design.

**Claims And Evidence:**

Yes, the claims made in the submission are supported by clear and convincing empirical results.

**Essential References Not Discussed:**

It would make the paper stronger if the authors could connect to ideas to improve deployment efficiency beyond FLOPs.

**Experimental Designs Or Analyses:**

Yes, I checked the experiment designs.

**Methods And Evaluation Criteria:**

Yes, the evaluation setup and results are standard for developing new scaling laws.

**Other Comments Or Suggestions:**

Line 82: Sentence is unfinished.

Line 217: Duplicated "and".

**Other Strengths And Weaknesses:**

Strengths:
1. The authors provide a comprehensive empirical analysis spanning multiple compute budgets and tasks, which provides new insights for MoE design.
2. The empirical analysis offers clear visualizations that effectively convey the trade-offs between total parameters, active parameters, and compute.

Weaknesses:
1. Sparsity improves deployment efficiency in production. It would make the paper stronger to discuss how the sparsity results can lead to real-world benefits beyond FLOPs where memory and communication costs matter. Also, it would make the paper stronger if connections could be made to other methods introducing sparsity to LLM, such as activation sparsity.

**Questions For Authors:**

See the above section.

**Relation To Broader Scientific Literature:**

This work extends existing scaling laws to MoE settings.

**Theoretical Claims:**

The paper is mainly empirical. The derivation of scaling laws was presented clearly.

---

> ### Author Rebuttal · Authors · 2025-03-28
>
> We thank the reviewer for their thoughtful and thorough review and for their support. We thank the reviewer for pointing out the comprehensive nature of our empirical work and presentation and respond to the question(s) raised by them below:
>
> ### Response to Weaknesses
>
> > Sparsity improves deployment efficiency in production. It would make the paper stronger to discuss how the sparsity results can lead to real-world benefits beyond FLOPs where memory and communication costs matter. Also, it would make the paper stronger if connections could be made to other methods introducing sparsity to LLM, such as activation sparsity.
>
> We thank the reviewer for bringing up activation sparsity. This is an interesting question as activation sparsity in language modeling is an active area of research [1, 2, 3]. We chose to focus on the most typical setup for inducing sparsity in MoEs, focusing mostly on model size and compute cost, and deferred studying other forms of sparsity that may have their own tradeoffs to future work. We once again thank the reviewer for bringing this interesting question to our attention.
>
> [1] Mirzadeh et al. Relu strikes back: Exploiting activation sparsity in large language models 2023
> [2] Szatkowski et al. Exploiting Activation Sparsity with Dense to Dynamic-k  Mixture-of-Experts Conversion 2024
> [3] Liu et al. TRAINING-FREE ACTIVATION SPARSITY IN LARGE LANGUAGE MODELS 2025
>
> ### Other comments
>
> > Line 82: Sentence is unfinished.
> > Line 217: Duplicated "and".
>
> We thank the reviewer for spotting these typos in our draft. We will fix these errors and carefully proofread to ensure we catch and fix other errors in the final version of our paper. We once again thank the reviewer for their review and support. We look forward to engage further with the reviewer if there are any additional questions.

---

### Official Review · Reviewer_uTJa · 2025-03-12

**Overall Recommendation:** 4

**Summary:**

This paper investigates the relationship between the number of model parameters and the compute per example, measured in Floating Point Operations (FLOPs), in the context of sparse Mixture-of-Experts (MoE) language models. The authors aim to understand how varying the sparsity level—defined as the fraction of inactive experts—affects model performance during pretraining and downstream tasks.

**Claims And Evidence:**

Yes

**Essential References Not Discussed:**

One possible discussion about the sparsity in the expert level in ACL2024.
[1]Not All Experts are Equal: Efficient Expert Pruning and Skipping for Mixture-of-Experts Large Language Models

**Experimental Designs Or Analyses:**

Yes

**Methods And Evaluation Criteria:**

Yes

**Other Comments Or Suggestions:**

No

**Other Strengths And Weaknesses:**

1. The paper is well written, I really enjoy reading the paper. This paper is interesting and important. The findings of this paper is important for future MOE LLM training and architecture designs.
2. The experiments are comprehensive and inspiring.

**Questions For Authors:**

1. Apart from the training loss, any scaling laws about some benchmark performance?
2. Is there any difference between training moe from scratch of continue pretrain from smaller dense model?

**Relation To Broader Scientific Literature:**

This paper further explores the scaling of MOE structured models.

**Theoretical Claims:**

No proofs in this paper

---

> ### Author Rebuttal · Authors · 2025-03-28
>
> We thank the reviewer for their thoughtful and thorough review and for their support.
>
> > Essential References Not Discussed
>
> The reference pointed to by the reviewer discusses techniques to sparsify MoEs after training whereas we discuss optimal sparsity during pretraining and its implications on downstream tasks. We will discuss the reference mentioned by the reviewer[1] in the related works section in the updated version of our paper as that can help the reader navigate the vast literature on MoEs. We thank the reviewer for bringing this reference to our attention.
>
> ### Response to Questions (Q)
>
> > Apart from the training loss, any scaling laws about some benchmark performance?
>
> Q1: We thank the reviewer for this question. Our experiments suggest that transferring performance from pretraining to few-shot downstream (DS) tasks depends on the nature of the task. We find this observation intriguing and believe that there is more work to be done to uncover the reasons behind this behavior, offering one hypothesis that tasks that can benefit strongly from more inference-time compute can be hampered by increasing sparsity naively.
> More generally, our trained models may not be suited for additional downstream evaluations since we do not post-train (RLHF, instruction finetuning) our models. We agree with the reviewer that the question of how sparsity affects DS evaluations but this work work is out of the scope of this paper but is an excellent topic for future work.
>
>
> > Is there any difference between training moe from scratch of continue pretrain from smaller dense model?
>
> We chose the most typical setup to study sparsity behavior in MoEs that uses multiple experts in the feed-forward network (FFN) layer.  Understanding the behaviors of pretraining starting from a smaller dense checkpoint, for e.g. MoEfication [1, 2] while interesting is outside the scope of this work. We can discuss this as future work though.
>
> [1] Shang et al. MoEfication: Transformer Feed-forward Layers are Mixtures of Experts 2022
> [2] Szatkowski et al. Exploiting Activation Sparsity with Dense to Dynamic-k  Mixture-of-Experts Conversion 2024
>
> We once again thank the reviewer for their review and support. We look forward to engage further with the reviewer if there are any additional questions.

---

### Official Review · Reviewer_r3sS · 2025-03-22

**Overall Recommendation:** 2

**Summary:**

The paper provides empirical scaling laws for MoE-based LLMs. The experimental setup is simply training MoE-LLMs on the RedPajama dataset and then evaluating by comparing the eval loss. With this setup the authors find:

1. For every sparsity level and FLOPs budget, there seems to be a unique optimal model size (fig. 1 and 2)
2. The optimal model sparsity increases as a function of total model size (fig. 4)
3. Downstream performance scale reliably w.r.t. validation loss irrespective of the model sparsity (fig. 5)

## update after rebuttal

I will keep the current score.

**Claims And Evidence:**

The claims themselves are reasonably well supported, but I'm not sure they are very novel.

**Essential References Not Discussed:**

No references missing AFAIK, but more details needed.

**Experimental Designs Or Analyses:**

Checked the experimental setup, seems reasonable.

**Methods And Evaluation Criteria:**

Yes, the methodology to evaluate the LLMs is sound. The scaling laws fits are evaluated by MSE which is not a good metric. It's better to use a scale-invariant metric like R^2.

**Other Comments Or Suggestions:**

1. Please give some more details on the experimental setup in the main paper. Currently it's hidden in the appendix.

**Other Strengths And Weaknesses:**

Pros:

1. The paper is well-written.
2. Scaling laws are impactful.


Cons:

1. The novelty is lacking. There have already been many scaling papers for MoEs and it's not clear what is new here. The paper shows that for a given FLOPs budget and sparsity there is an optimal model size. This has been known for dense models for a long time, and one would expect it to hold for MoE models too.

2. The evaluation of the fits is lacking. The authors evaluate that with MSE, it's better to use a scale-invariant metric like R^2.

3. It is not clear if there are any new lessons for practitioners. Everyone knows that MoEs work well and that balancing training duration and model size is needed.

**Questions For Authors:**

1. Can you provide R^2 metrics for the fits and evaluate them on extrapolation?
2. Why do you use “scale-free Adam optimizer”? It is non-standard.
3. Regarding related work you write “However, these studies typically assume a fixed configuration for other critical variables influencing FLOPs per token, such as the number of active experts per input.”  -- could you give more details on exactly what previous studies cover, and how it differs from what you cover?
4. Fig 1 says “These results indicate that for a fixed compute budget, increasing model sparsity leads to a reduction in pretraining loss” — this seems to not be true e.g. in figure 4. Please clarify this.

**Relation To Broader Scientific Literature:**

na

**Theoretical Claims:**

Na

---

> ### Author Rebuttal · Authors · 2025-03-28
>
> ### Response to Weaknesses (W):
>
> > The novelty is lacking. There have already been many scaling papers for MoEs and it's not clear what is new here. The paper shows that for a given FLOPs budget and sparsity there is an optimal model size. This has been known for dense models for a long time, and one would expect it to hold for MoE models too.
>
> W1: We thank the reviewer for raising this point about prior work on dense models that relates optimal model size for a given FLOPs budget. Our intention with this work is to shed  light on how model size and sparsity jointly interact for compute-optimal models. While the reviewer correctly points out that one could hypothesize that MoEs should scale in an analogous way to dense models based on prior studies by setting sparsity to a fixed value, it is unclear how one might set this sparsity value optimally. We further study the practical case of fixed model size, which can be particularly important in on-device deployment scenarios, and suggest a parametric form for scaling laws that the reviewer highlighted as a strength of the paper.
> In general, our study suggest that when designing a new architecture, or hardware, we should keep in mind that allowing the model to have more parameters vs more FLOPs per example would be more efficient as long as the goal is to do well on the pre-training task and DS tasks that are well correlated with the pre-training performance of the model.
>
> > The evaluation of the fits is lacking. The authors evaluate that with MSE, it's better to use a scale-invariant metric like R^2.
>
> W2: The R^2 values are 99% on fitting data  and 68% on the held out extrapolation dataset (sparsity = 98%). We thank the reviewer for raising this point. We will update our paper with the above results.
>
> > It is not clear if there are any new lessons for practitioners. Everyone knows that MoEs work well and that balancing training duration and model size is needed.
>
> W3: The findings from this study suggest that MoEs don’t always work well unless sparsity values are set carefully. Conceretely, we find that optimal sparsity values grows with model size. Additionally, we find that MoEs transfer performance depends on the nature of the downstream task (knowledge vs reasoning) which is consistent with the observations made by in concurrent work by Jelassi et al.[1]. These are valuable findings to both practitioners & researchers that want to build efficient MoEs.
>
> [1] Jelassi et al. Mixture of Parrots: Experts improve memorization more than reasoning.  October 2024 / ICLR 2025
>
> ### Response to Questions (Q)
>
> > Can you provide R^2 metrics for the fits and evaluate them on extrapolation
> Q1: We provide R^2 metrics in our response to W2 above.
>
> > Why do you use “scale-free Adam optimizer”? It is non-standard.
> Q2: We thank the reviewer for carefully reading our paper and appendix. We used AdamW described in Loshchilov and Hutter (https://arxiv.org/abs/1711.05101). We will clarify this detail in the final version.
>
> > Regarding related work you write “However, these studies typically assume a fixed configuration for other critical variables influencing FLOPs per token, such as the number of active experts per input.” — could you give more details on exactly what previous studies cover, and how it differs from what you cover?
> Q3: We highlight the important papers to answer the reviewer’s question
> - Clark et al. assume a fixed size dataset of 130 billion tokens to derive scaling laws.
> - Ludziejewski & Krajewski et al. conduct their experimetns at a much smaller scale compared to our work and focus on varying granularity while we also consider number of active experts per input and using expert-choice routing while we use token-choice routing
>
> We have discussed related work more extensively in Appendix A. We will gladly discuss additional works that the reviewer may want us to do so.
>  We thank the reviewer for asking this clarification and are happy to provide further information.
>
> > Fig 1 says “These results indicate that for a fixed compute budget, increasing model sparsity leads to a reduction in pretraining loss” — this seems to not be true e.g. in figure 4. Please clarify this.
>
> Q4: Figure 4 studies the case where the model size is a constraint and shows how sparsity value changes for a given model size. If there is no bound on the total number of parameters then optimal sparsity level approaches 1. OTOH Figure 1 shows that for a given compute budget, optimal models with higher sparsity have a larger parameter count and smaller active parameter count. We thank the reviewer for this question and will adjust the captions to make things more clear in our final version.
>
> We thank the reviewer for carefully reading our paper and the appendix! We will follow this suggestion and promise to update the draft in the final version of our paper. If our rebuttal is satisfactory, we ask the reviewer to consider raising their score.

---

### Decision · Program_Chairs · 2025-05-01

**Decision:**

Accept (poster)

**Comment:**

The paper studied the trade-off between sparsity and performance in MoE models under different constraints. The author found that given A fix parameter size or total training compute, there is an optimal level of sparsity that improves both training efficiency and model performance.

Reviewers generally agree that the research topic is important, and this paper is well-written and the experiment setting is comprehensive.